# MathGLM-Vision: Solving Mathematical Problems with Multi-Modal Large Language Model

## Abstract

Large language models (LLMs) have demonstrated significant capabilities in mathematical reasoning, particularly with text-based mathematical problems. However, current multi-modal large language models (MLLMs), especially those specialized in mathematics, tend to focus predominantly on solving geometric problems but ignore the diversity of visual information available in other areas of mathematics. Moreover, the geometric information for these specialized mathematical MLLMs is derived from several public datasets, which are typically limited in diversity and complexity. To address these limitations, we aim to construct a fine-tuning dataset named MathVL, and develop a series of specialized mathematical MLLMs termed MathGLM-Vision by conducting Supervised Fine-Tuning (SFT) on MathVL with various parameter-scale backbones. To extensively evaluate the effectiveness of MathGLM-Vision, we conduct experiments on several public benchmarks and our curated MathVL-test benchmark consisting of 2,000 problems. Experimental results demonstrate that MathGLM-Vision achieves significant improvements compared with some existing models, including backbone models and open-source mathematical MLLMs. These findings indicate the importance of diversity dataset in enhancing the mathematical reasoning abilities of MLLMs. Both MathGLM-Vision model (based on CogVLM2, GLM-4V-9B) and MathVL-test will be open-sourced.

## 1 Introduction

Recent advancements in computational linguistics have led to substantial progress in solving mathematical problems using Large Language Models (LLMs) with multi-step reasoning processes (Lightman et al., 2023). For example, models like GPT-4 (Achiam et al., 2023), Qwen (Bai et al., 2023a), GLM-4 (Team et al., 2024), LLaMA (Touvron et al., 2023a;b) have demonstrated impressive performance on mathematical datasets such as GSM8K (Cobbe et al., 2021) and MATH (Hendrycks et al., 2021). Furthermore, the development of specialized mathematical models is expanding the potential of LLMs in this domain. These models, specifically designed for mathematical problem solving, include notable contributions such as WizardMath (Luo et al., 2023), MAmmoTH (Yue et al., 2023), MathCoder (Wang et al., 2023a), MetaMath (Yu et al., 2023), DeepSeekMath (Shao et al., 2024), and others (Yang et al., 2023; Yuan et al., 2023; Gou et al., 2023; Yue et al., 2024b; Mitra et al., 2024; Ying et al., 2024). These advancements highlight the growing proficiency of LLMs in handling intricate mathematical reasoning and problem-solving tasks.

Despite significant advancements, the majority of models designed for mathematical problem solving still rely predominately on textual representations. This limits their effectiveness in scenarios that require visual information. Notably, approximately 63% of mathematics questions in Chinese K12 education include visual elements, highlighting the critical role of visual information in comprehending and solving mathematical problems.

Therefore, a crucial question arises: **Is visual information essential for solving these mathematical problems that include visual elements?** To verify this, we conduct a series of insightful experiments comparing the performance of these models such as GPT-4o, Claude-3.5-Sonnet, Qwen-VL-Max, and Gemini-1.5-Pro on MathVL-test dataset, both with and without visual inputs. As shown in Figure 1, the results clearly demonstrate that the inclusion of visual elements significantly enhances the models' ability to accurately solve complex mathematical problems. Conversely, the exclusion of

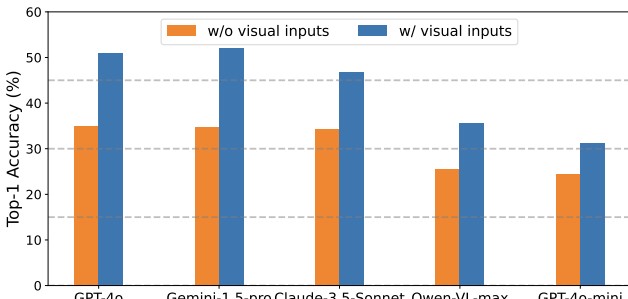 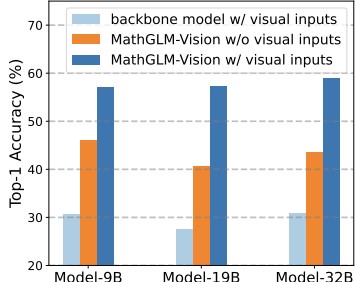

Figure 1: Insight experiments demonstrates the significance of visual information in solving mathematical problems. (Left) A performance comparison of different models with and without visual inputs on MathVL-test dataset. (Right) The accuracies of MathGLM-Vision on MathVL-test with and without visual inputs.

visual information leads to a pronounced decrease in performance, emphasizing the essential role that visual context plays in solving mathematical problems that incorporate visual elements.

Currently, multi-modal large language models (MLLMs) are at the forefront of efforts to integrate visual and textual information for solving mathematical problems. Close-source models such as GPT-4V (OpenAI, 2023), Gemini (Team et al., 2023), Claude3 (Anthropic, 2024), Qwen-VL (Bai et al., 2023b), along with several open-source MLLMs like CogVLM (Wang et al., 2023c; Hong et al., 2024), MiniGPT Zhu et al. (2023), LLaVA-1.5 Liu et al. (2024a), SPHINX-MoE Gao et al. (2024), and LLaVA-NeXT Liu et al., demonstrate substantial potential in addressing geometric reasoning challenges. Additionally, specialized geometric MLLMs like G-LLaVA Gao et al. (2023a), GeoGPT4V Cai et al. (2024) and Math-LLaVA Shi et al. (2024) are particularly focused on enhancing capabilities in this domain. However, these models still face several challenges and limitations that need to be addressed.

- *Current MLLMs, particularly those specialized in mathematics, predominantly focus on solving geometric problems and tend to overlook the diversity of visual information in mathematics. This visual information encompasses a broad spectrum of elements, including arithmetic, statistics, algebra and word problems, each integral to different mathematical domains beyond geometry.*

- *Current fine-tuning dataset for specialized mathematical MLLMs, typically sourced from public datasets like GeoQA and Geometry3K, often lack diversity and complexity. This limitation restricts the models' ability to effectively solve a broader range of mathematical problems.*

- *Current specialized mathematical MLLMs are predominantly designed to process single-image inputs and lack the capability to handle multiple images simultaneously. This limitation hampers their ability to tackle complex problems that necessitate the integration of information from multiple visual sources.*

In response to these challenges and limitations, we construct a fine-tuning dataset named **MathVL**, which encompasses both open-source data and our specially curated Chinese data collected from K12 education. The MathVL dataset is meticulously designed to incorporate a diverse range of mathematical problems, consisting of textual and visual inputs. For textual information, the MathVL dataset covers a variety of mathematical subjects such as arithmetic, algebra, geometry, statistics, and word problems. It includes various types of questions, including fill-in-the-blank, multiple-choice, and free-form. For visual information, the MathVL dataset involves elements like functions, statistical data, graphs, charts, LaTeX expressions, and geometric figures, providing a comprehensive resource for complex mathematical problem solving.

With our constructed MathVL dataset, we develop a series of specialized mathematical MLLMs, collectively referred to as **MathGLM-Vision**, with different parameter scales. Specifically, MathGLM-Vision-9B, MathGLM-Vision-19B and MathGLM-Vision-32B are fine-tuned on three backbone models: GLM-4V-9B, CogVLM2, and CogVLM-32B, respectively. Moreover, we establish a benchmark dataset named **MathVL-test,** which contains 2,000 problems designed to evaluate the ability of MathGLM-Vision and other MLLMs in solving mathematical problems involving multiple images. Through extensive evaluation experiments on three public benchmark datasets and one curated

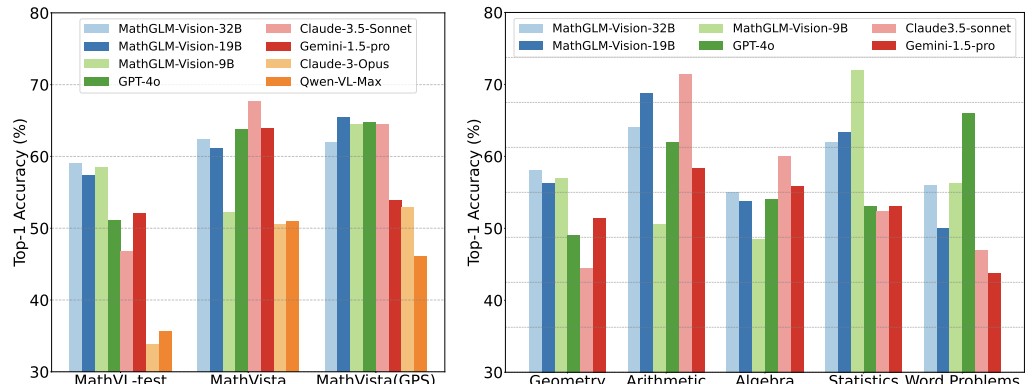

Figure 2: Performance comparison of the different multi-modal large language models. (Left) The accuracies of MathGLM-Vision and other MLLMs among three evaluation datasets. (Right) The accuracy of MathGLM-Vision and other MLLMs on MathVL-test across different categories.

MathVL-test, we validate the effectiveness of our MathGLM-Vision. The results in Figure 2 demonstrate that MathGLM-Vision exhibits superior performance in understanding and solving complex mathematical problems with visual elements compared to existing MLLMs. For instance, on the geometry problem solving (GPS) minitest split of MathVista (Lu et al., 2023), MathGLM-Vision-9B achieves a 39.68% relative improvement for GLM-4V-9B, MathGLM-Vision-19B achieves a 65.06% relative improvement for CogVLM2, and MathGLM-Vision-32B achieves a 51.05% relative improvement over CogVLM-32B. Last but not least, Both MathGLM-Vision model (based on CogVLM2, GLM-4V-9B) and MathVL-test will be open-sourced to facilitate the future development of this field.

We highlight our contributions as follows:

- **Data Perspective:** We construct MathVL, a diverse and comprehensive multi-modal mathematical supervised fine-tuning dataset that contains both textual and visual inputs.

- **Model Perspective:** We develop a suite of specialized mathematical multi-modal large language models, referred to as MathGLM-Vision, which demonstrates significant improvements on various mathematical benchmarks while maintaining general vision-language understanding capabilities.

- **Benchmark Perspective:** We establish a benchmark dataset called MathVL-test, which designed to evaluate the mathematical reasoning abilities of MLLMs using a multi-image format.

## 2  MATHVL: DATASET CURATION

To enhance the capabilities of MLLMs in solving mathematical problems, previous efforts (Chen et al., 2021; 2022; Cao & Xiao, 2022; Gao et al., 2023a) focus on constructing high-quality datasets. Nevertheless, the majority of these datasets fall into the category of Visual Question Answering (VQA), which generally involves descriptive or identification tasks rather than conventional mathematical problems. Furthermore, the answers in some public datasets like Geometry3K (Lu et al., 2021), GeoGPT4V (Cai et al., 2024), MathV360K (Shi et al., 2024) for standard mathematical questions are often too simplistic, usually providing only the final answer without the intermediate steps necessary for a thorough understanding. It is well-established that including step-by-step solutions can significantly enhance the reasoning capabilities of large language models (Wei et al., 2022; Lightman et al., 2023; Zhang et al., 2023; Wang et al., 2023b). Figure 3 demonstrates the distribution of answer lengths in current open-source mathematical datasets.

To address these issues, we construct a fine-tuning dataset MathVL, including both several public datasets and our curated Chinese dataset collected from K12 education levels. This dataset is meticulously crafted to encompass a diverse array of mathematical problems that incorporate visual information. Each problem is presented with detailed step-by-step solutions, aiming to enhance the problem-solving skills of MLLMs by providing them with both the context and the procedural knowledge necessary for effective reasoning and comprehension.

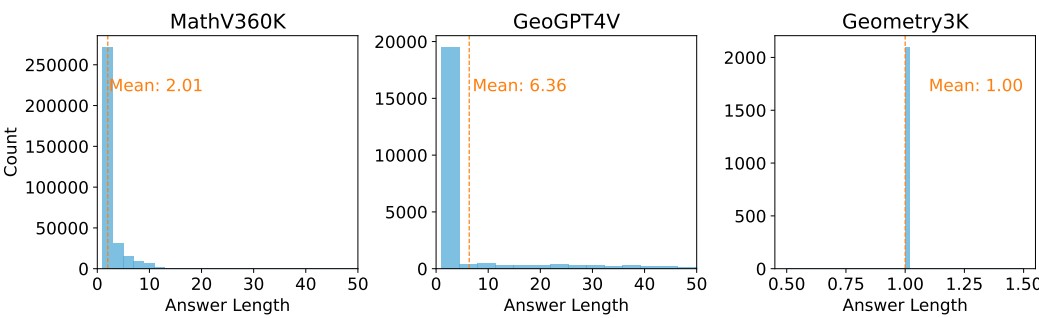

Figure 3: Analysis of answer lengths in several open-source mathematical datasets like MathV360K, GeoGPT4V, and Geometry3K.

**Open-Source Data.** We first collect open-source datasets from GeoQA+ (Cao & Xiao, 2022), Geometry3K (Lu et al., 2021), ChartQA (Masry et al., 2022), and UniGEO-Calculation (Chen et al., 2022). These datasets commonly serve as seed data for constructing enhanced datasets. Through observation and statistical analysis, we discover that 57% of the answers within these datasets are comprised of fewer than 50 words, indicating that many questions are answered directly without elaboration or explanation. To enrich these dataset with comprehensive step-by-step solutions, we employ GPT-4o to generate the detailed solutions for each question, thereby enhancing the learning and reasoning potential of these datasets. After generating the detailed answers, we perform a rigorous judgement process to ensure the accuracy of the solutions provided by GPT-4o. Additionally, we adopt a public instruction tuning dataset named Geo170K (Gao et al., 2023a), which is constructed using GeoQA+ and Geometry3K as seed data and contains more than 110K geometric question-answer pairs. We also incorporate another public dataset, GeomVerse (Kazemi et al., 2023), as part of our resources. In the end, the detailed statistics of the open-source datasets used in MathGLM-Vision is provided in Table 1.

| Datasets | ChartQA | UniGeo-Calculation | Geometry3K | GeoQA+ | Geo170K | GeomVerse | ALL |
|---|---|---|---|---|---|---|---|
| Samples | 7,398 | 3,499 | 2,101 | 6,026 | 117,205 | 9,339 | 145,568 |

Table 1: The detailed statistics of the open-source datasets used in MathGLM-Vision.

**Chinese Data Collected from K12 Education.** We construct a dataset specifically focused on K12 education, comprising 341,346 mathematical problems with textual and visual inputs. This dataset is meticulously curated to encompass a board range of mathematical topics and difficulty levels tailored to the Chinese educational curriculum. It features various question types, such as multiple-choice, fill-in-the-blank, and free-form questions, spanning disciplines including arithmetic, algebra, geometry, statistics, and word problems. Mathematically, this dataset can be represented as $D_{\text{MathVL}}^{\text{zh}} = \{Q, A, I_s\}$, where $Q$ represents the question, $A$ represents the answer, and $I_s$ represents one or more images associated with each question. To build this dataset, we first process the images by adding a white border around each image and enhancing their resolution to ensure that MLLMs can effectively recognize and interpret these images. This modification is crucial for facilitating the accurate extraction of visual information. Next, we extract 341,346 samples from a raw dataset containing 685,670 samples by implementing a selective filtering process. This selection is based on two specific criteria: (1) filtering out samples where the answer includes images or the question is incomplete, and (2) eliminating samples with answer that are fewer than 50 words in length to ensure the responses are sufficiently detailed for model training. After constructing this dataset, we categorize and analyze it based on mathematical topics associated with each question. Detailed statistics about the distribution of these categories are presented in Table 2. Figure 4 demonstrates some examples sampled from the constructed Chinese dataset, providing a visual representation of the mathematical topics of questions included. More dataset cases are provided in Appendix A.

| Types | Arithmetic | Geometry | Algebra | Statistics | Word Problems | ALL |
|---|---|---|---|---|---|---|
| Samples | 7,207 | 291,879 | 20,111 | 18,284 | 3,865 | 341,346 |

Table 2: Detailed statistics regarding the distribution used in MathGLM-Vision.

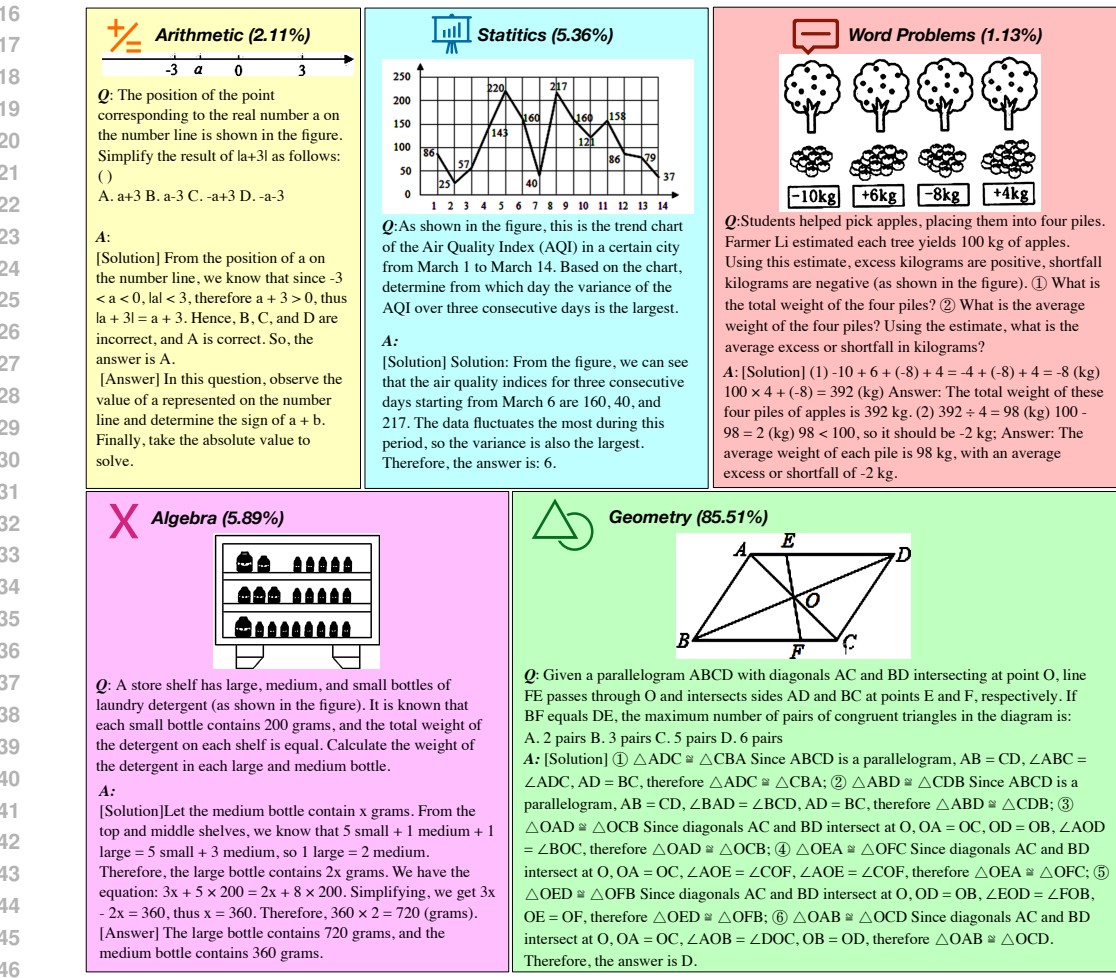

Figure 4: Examples sampled from the constructed Chinese dataset.

## 3 MATHGLM-VISION: MODEL TRAINING

**Model Architecture.** We employ CogVLM2 (Hong et al., 2024; Wang et al., 2023c) and GLM-4V-9B (GLM et al., 2024) architectures as our backbone models, and conduct Supervised Fine-Tuning (SFT) on our constructed MathVL dataset. Specifically, we utilize three pre-trained multi-modal large language models (MLLMs) for the fine-tuning process: GLM-4V-9B, CogVLM2-19B, and CogVLM-32B. This results in the development of three distinct variants of MathGLM-Vision, designated as MathGLM-Vision-9B, MathGLM-Vision-19B, and MathGLM-Vision-32B, respectively. Further details about the abovementioned three pre-trained MLLMs are available in Appendix B.

**Model Training.** To maintain the general vision-language understanding skills of MathGLM-Vision, we incorporate 19 open-source visual question-answering datasets (VQA datasets) into the MathVL dataset. More details about the task type and visual context of VQA datasets are provided in Appendix C. These datasets are meticulously selected to challenge and enhance the model's ability to interpret and integrate visual and textual information, ensuring it retains a broad understanding across various contexts. By merging these varied sources, we enhance MathGLM-Vision's specialized capabilities for mathematical problem-solving and simultaneously preserve its robustness in general vision-language tasks. In the end, we conduct supervised fine-tuning (SFT) across the combined VQA and MathVL datasets. The training process undergoes 35,000 iterations with a learning rate of 1e-5 and a batch size of 128. To ensure the stability of the training, we activate the visual encoder's parameters and adjust its learning rate to be one-tenth of that used for the remaining training parameters. The details of the SFT procedures are described in Appendix D.

## 4 EXPERIMENTS

### 4.1 EXPERIMENTAL SETUP

**Evaluation Datasets.** We assess our MathGLM-Vision using three well-established public benchmark datasets (MathVista (Lu et al., 2023), MathVerse (Zhang et al., 2024), and MATH-Vision (Wang et al., 2024) datasets) alongside our specially curated dataset MathVL-test benchmark. This benchmark comprises 2,000 sampled cases, distinct from those in the MathVL dataset, ensuring a rigorous and unbiased evaluation of MathGLM-Vision's capabilities. Additionally, we evaluate MathGLM-Vision's general vision-language understanding skills using the MMMU benchmark Yue et al. (2024a). Detailed descriptions for these benchmark datasets are provided in Appendix E.

**Compared Models.** We compare MathGLM-Vision with other Multi-Modal Large Language Models (MLLMs), including closed-source MLLMs such as Gemini (Team et al., 2023), GPT-4V (OpenAI, 2023), Claude3 (Anthropic, 2024), and Qwen-VL (Bai et al., 2023b), and open-source MLLMs like mPLUG-Owl (Ye et al., 2023), LLaMA-Adapter-V2 (Gao et al., 2023b), InstrctBLIP (Dai et al., 2024), LLaVA-1.5 (Liu et al.), ShareGPT4V (Chen et al., 2023), SPHINX (Gao et al., 2024), InternLM-XC2 (Dong et al., 2024), and InternVL (Chen et al., 2024). Additionally, we compare MathGLM-Vision with recent specialized mathematical MLLMs, including G-LLaVA (Gao et al., 2023a), LLaVA-1.5-G (Cai et al., 2024), ShareGPT4V-G (Cai et al., 2024), and Math-LLaVA (Shi et al., 2024).

**Evaluation Metrics.** We adopt top-1 accuracy to evaluate our MathGLM-Vision across MathVista-GPS, MathVista, MathVerse, MATH-V, and MathVL-test benchmarks. Our evaluation process follows the pipeline outlined in the aforementioned benchmark datasets, which involves using LLMs to extract predicted answers from the model's responses. Accuracy is then calculated by comparing these extracted answers against the ground truths.

### 4.2 MAIN RESULTS

**Results on public benchmark datasets.** To comprehensively assess the ability of MathGLM-Vision in solving mathematical problems, we evaluate its performance against other MLLMs across several public benchmark datasets. Table 3 demonstrates the overall results from these evaluations. The experimental results indicate that our constructed MathVL dataset can significantly improve MathGLM-Vision's mathematical reasoning capabilities. For example, MathGLM-Vision-9B achieves a 64.42% accuracy on the MathVista-GPS dataset, marking a substantial 39.68% improvement over its backbone model, GLM-4V-9B. Besides, across various parameter scales, MathGLM-Vision consistently surpass all backbone models on different evaluation benchmarks, highlighting the significant enhancements that MathVL brings to the MathGLM-Vision's problem-solving skills. Notably, MathGLM-Vision outperforms all open-source specialized mathematical MLLMs across various benchmarks. The superior performance suggests that the high-quality and diverse data, complete with detailed step-by-step solutions, are crucial for improving MLLM's mathematical reasoning capabilities. More importantly, MathGLM-Vision-32B outperforms even the advanced GPT-4V on the more challenging MATH-V benchmark, demonstrating its superior capacity to tackle complex mathematical problems. Detailed experimental results on public benchmark datasets across different task types can be found in Appendix F.

**Results on MathVL-test.** We also evaluate MathGLM-Vision and several close-source MLLMs using our specially constructed MathVL-test benchmark. As depicted in Table 4, the results clearly demonstrate that MathGLM-Vision significantly outperforms both its backbone models and other leading closed-source MLLMs across various model sizes. Specifically, our MathGLM-Vision-32B outperforms the advanced GPT-4o with a significant margin, achieving an accuracy of 59.00% compared to GPT-4o's 51.05%. Compared to the backbone model, GLM-4V-9B, MathGLM-Vision-9B achieves an impressive accuracy of 57.05% with a significant improvement of 86.5%. This superior performance suggests that MathGLM-Vision, when conducting SFT on the MathVL dataset, notably enhances its capability to tackle complex Chinese mathematical problems. Additionally, we report the accuracy across various categories, as illustrated in Figure 2 (See Right). MathGLM-Vision significantly outperforms other advanced MLLMs in the domains of geometry and statistics. In

| Model | Input | LLM | MathVista (GPS) | MathVista | MathVerse | MATH-V |
|---|---|---|---|---|---|---|
| *Closed Source Models* | | | | | | |
| Gemini Pro | $Q, I$ | - | 40.40 | 45.20 | 36.80 | 17.66 |
| Gemini-1.5-Pro | $Q, I$ | - | 53.85 | 63.90 | 51.08 | 19.24 |
| GPT-4V | $Q, I$ | - | 50.50 | 49.90 | 50.80 | 22.76 |
| GPT-4-turbo | $Q, I$ | - | 58.25 | 58.10 | 43.50 | 30.26 |
| GPT-4o | $Q, I$ | - | 64.71 | 63.80 | 56.65 | 30.39 |
| Claude3-Opus | $Q, I$ | - | 52.91 | 50.50 | 31.77 | 27.13 |
| Claude3.5-Sonnet | $Q, I$ | - | 64.42 | 67.70 | 48.98 | 37.99 |
| Qwen-VL-Plus | $Q, I$ | - | 33.01 | 43.30 | 19.10 | 10.72 |
| Qwen-VL-Max | $Q, I$ | - | 46.12 | 51.00 | 35.90 | 15.59 |
| *Open Source Models* | | | | | | |
| *General Multi-modal LLMs* | | | | | | |
| mPLUG-Owl | $Q, I$ | LLaMA-7B | 23.60 | 22.20 | 12.47 | 9.84 |
| LLaMA-Adapter-V2 | $Q, I$ | LLaMA-7B | 25.50 | 23.90 | 4.50 | 9.44 |
| InstructBLIP | $Q, I$ | Vicuna-7B | 20.70 | 25.30 | 15.36 | 10.12 |
| LLaVA-1.5 | $Q, I$ | Vicuna-13B | 24.04 | 27.60 | 12.70 | 11.12 |
| ShareGPT4V | $Q, I$ | Vicuna-13B | 38.35 | 29.30 | 16.20 | 11.88 |
| SPHINX-MoE | $Q, I$ | Mixtral 8*7B | 31.20 | 42.30 | 19.60 | 14.18 |
| SPHINX-Plus | $Q, I$ | LLaMA2-13B | 16.40 | 36.70 | 14.70 | 9.70 |
| InternLM-XC2 | $Q, I$ | InternLM2-7B | 63.00 | 57.60 | 24.40 | 14.54 |
| InternVL-1.2-Plus | $Q, I$ | Nous-Hermes-2-Yi-34B | 61.10 | 59.90 | 21.70 | 16.97 |
| *Geo-Multi-modal LLMs* | | | | | | |
| G-LLaVA | $Q, I$ | LLaMA2-7B | 53.40 | 28.46 | 12.70 | 12.07 |
| G-LLaVA | $Q, I$ | LLaMA2-13B | 56.70 | 35.84 | 14.59 | 13.27 |
| LLaVA-1.5-G | $Q, I$ | Vicuna-7B | 32.69 | 45.22 | 13.96 | 14.13 |
| LLaVA-1.5-G | $Q, I$ | Vicuna-13B | 36.54 | 48.34 | 15.61 | 14.88 |
| ShareGPT4V-G | $Q, I$ | Vicuna-7B | 32.69 | 45.07 | 16.24 | 12.86 |
| ShareGPT4V-G | $Q, I$ | Vicuna-13B | 43.27 | 49.14 | 16.37 | 14.45 |
| Math-LLaVA | $Q, I$ | Vicuna-13B | 57.70 | 46.60 | 19.04 | 15.69 |
| *MathGLM-Vision and Backbone Models* | | | | | | |
| GLM-4V-9B | $Q, I$ | GLM-4-9B | 46.12 | 46.70 | 35.66 | 15.31 |
| **MathGLM-Vision-9B** | $Q, I$ | GLM-4-9B | **64.42** | **52.20** | **44.20** | **19.18** |
| CogVLM2 | $Q, I$ | LLaMA-3-8B | 39.61 | 40.85 | 25.76 | 13.20 |
| **MathGLM-Vision-19B** | $Q, I$ | LLaMA-3-8B | **65.38** | **61.10** | **42.50** | **21.64** |
| CogVLM-32B | $Q, I$ | GLM2-32B | 41.06 | 40.04 | 35.28 | 19.32 |
| **MathGLM-Vision-32B** | $Q, I$ | GLM2-32B | **62.02** | **62.40** | **49.20** | **26.51** |

Table 3: **Results on several public benchmark datasets.** Comparison of model performance on the testmini set of MathVista and geometry problem solving (GPS) of MathVista. For MathVerse dataset, results are evaluated on Vision Dominant with CoT-E. For MATH-V dataset, all 3,040 samples included in the data are evaluated.

contrast, Claude3.5-Sonnet excels in algebra and arithmetic, demonstrating superior performance. Meanwhile, MathGLM-Vision-19B ranks second in performance in the domain of arithmetic, showing its strong abilities in this area as well. GPT-4o exhibits the highest performance in word problems domain, while MathGLM-Vision also exhibits robust performance, surpassing both Gemini-1.5-Pro and Claude3.5-Sonnet in this category.

## 4.3 GENERALIZABILITY OF MATHGLM-VISION

In addition to its proficiency in mathematical reasoning, we further assess MathGLM-Vision's capabilities in general vision-language understanding by conducting experiments on the MMMU benchmark. This benchmark is specifically designed to evaluate the ability of models to comprehend and process information across a variety of academic and professional disciplines, providing a comprehensive test of general vision-language understanding. Table 5 shows the performance of MathGLM-Vision, a specific variant fine-tuned exclusively on MathVL without the inclusion of VQA datasets, and backbone models. Compared to CogVLM2, MathGLM-Vision-19B achieves comparable performance in terms of generalizability, underscoring its capacity for simultaneous

| Model | Input | LLM Size | MathVL-test |
|---|---|---|---|
| Gemini-1.5-Pro | $Q, I$ | - | 52.03 |
| GPT-4V | $Q, I$ | - | 35.89 |
| GPT-4-turbo | $Q, I$ | - | 42.19 |
| GPT-4o | $Q, I$ | - | 51.05 |
| Claude3.5-Sonet | $Q, I$ | - | 46.84 |
| Claude3-Opus | $Q, I$ | - | 33.77 |
| Qwen-VL-Plus | $Q, I$ | - | 28.50 |
| Qwen-VL-Max | $Q, I$ | - | 35.61 |
| GLM-4V-9B | $Q, I$ | 9B | 30.59 |
| **MathGLM-Vision-9B** | $Q, I$ | 9B | **57.05** |
| CogVLM2 | $Q, I$ | 8B | 27.47 |
| **MathGLM-Vision-19B** | $Q, I$ | 8B | **57.30** |
| CogVLM-32B | $Q, I$ | 32B | 30.86 |
| **MathGLM-Vision-32B** | $Q, I$ | 32B | **59.00** |

Table 4: **Results on MathVL-test.** A detailed comparison of the performance of MathGLM-Vision and various other leading close-source MLLMs on the MathVL-test benchmark.

multi-modal understanding and mathematical reasoning. However, MathGLM-Vision-32B shows a slight reduction in performance across multiple categories on the MMMU benchmark. Besides, MathGLM-Vision, when fine-tuned with VQA datasets, outperforms its variant lacking VQA datasets. This indicates that omitting VQA datasets from the fine-tuning process limits the general vision-language understanding abilities. Thus, the SFT process using our MathVL incorporated with VQA datasets not only enhances MathGLM-Vision's mathematical reasoning abilities but also preserves its generalizability.

| Model | MMMU | Art & Design | Business | Sci. | Health & Med. | Human. & Social Sci. | Tech. & Eng. |
|---|---|---|---|---|---|---|---|
| CogVLM2 | 40.2 | 58.3 | 30.0 | 26.7 | **41.3** | 38.6 | **53.3** |
| w/o VQA datasets | 38.1 | 60.8 | 28.7 | **34.0** | 36.7 | 43.3 | 32.9 |
| MathGLM-Vision-19B | **40.2** | **63.3** | **37.3** | 27.3 | 36.0 | **46.7** | 37.6 |
| CogVLM-32B | **42.9** | **63.3** | 31.3 | 32.0 | **43.3** | 62.5 | 36.7 |
| w/o VQA datasets | 38.6 | 62.5 | 26.7 | 28.0 | 34.0 | 56.7 | 33.8 |
| MathGLM-Vision-32B | 40.0 | 60.0 | 28.7 | **34.0** | 38.7 | 52.5 | 34.8 |

Table 5: Generalizability of MathGLM-Vision on the MMMU benchmark.

## 4.4 FURTHER ANALYSIS

**Effect of Chinese Dataset.** To validate the effectiveness of the adopted Chinese dataset in MathVL, we conduct an extended experiment that involves fine-tuning GLM-4V-9B with open-source datasets, deliberately excluding Chinese data collected from K12 education. Table 6 shows a comparison of performance results. Compared to the backbone model GLM-4V-9B, a variant MathGLM-Vision-9B that undergoes SFT exclusively with open-source data exhibits significant improvement on the minitest of MathVista, particularly in geometry problem solving (GPS) and geometry reasoning (GEO). This indicates that fine-tuning on diverse open-source data can markedly enhance model performance in specific mathematical areas. MathGLM-Vision, incorporating both open-source data and Chinese data, outperforms the variant tuned only with open-source data on the minitest of MathVista, highlighting the significant value added by integrating the Chinese dataset in the training process. Notably, compared to the variant without Chinese data, MathGLM-Vision achieves a significantly higher accuracy on the MathVL-test benchmark. These findings confirm that the inclusion of the Chinese dataset not only enhances the model's capability in handling complex mathematical problems but also contributes significantly to its overall performance on a diverse set of tasks within MathVista.

**Effect of VQA Datasets.** To explore the effect of VQA datasets on the performance of MathGLM-Vision, an extended experiment can be designed where SFT is applied exclusively to mathematical

| Model | MathVista GPS GEO ALL | MathVL-test |
|---|---|---|
| GLM-4V-9B | 46.12 44.35 46.70 | 30.59 |
| + SFT on Open-source Data | 62.98 61.51 50.40 | 47.55 |
| MathGLM-Vision-9B | 64.42 62.34 52.20 | 57.25 |

Table 6: Effect of the constructed Chinese data.

| Model | MathVista GPS GEO ALL |
|---|---|
| GLM-4V-9B | 46.12 44.35 46.70 |
| MathGLM-Vision-9B | 64.42 62.34 52.20 |
| - SFT on VQA Datasets | 61.54 58.58 41.34 |

Table 7: Effect of the VQA datasets.

datasets, deliberately excluding VQA datasets. Table 7 demonstrates the performance comparison achieved by different models on MathVista. Compared to the backbone model GLM-4V-9B, a variant of MathGLM-Vision-9B achieves significant improvements on geometry problem solving (GPS) and geometry Reasoning (GEO). However, it exhibits a decline in the overall accuracy on the minitest of MathVista (ALL). The decline can be attributed to the composition of MathVista, which comprises five tasks, with question-answering types (such as graphical question-answering, textbook question-answering, and visual question-answering) comprising up to 60.6% of the tasks. Omitting VQA training in MathGLM-Vision impacts the model's ability to effectively process and respond to these multi-modal questions. Notably, within specific subsets of MathVista, such as GPS and GEO, a variant of MathGLM-Vision-9B slightly below the standard MathGLM-Vision-9B. This observation suggests that VQA datasets are crucial for preserving overall multi-modal understanding, their impact may vary depending on different task types. Besides, VQA datasets can indirectly bolster mathematical reasoning skills, which in turn enhances image recognition capabilities.

### 4.5 ERROR ANALYSIS

We meticulously analyze the causes of errors in MathGLM-Vision-32B on the MathVL-test benchmark and illustrate the distribution of these errors in Figure 5. We summarize these errors in MathGLM-Vision-32B into five types: reasoning error, knowledge error, vision recognition error, calculation error, and question misunderstood error. The most prevalent type of errors, accounting for 69.1% of the total, is identified as Reasoning Error. This indicates a significant challenge in the MathGLM-Vision-32B's logical deductions and inferential reasoning. Improving these capabilities can dramatically enhance the MathGLM-Vision-32B's overall performance. Knowledge Error, which made up 12.7% of the errors, relates to the model's misapplicaion or lack of specific factual information. Vision Recognition Error accounts for 11.4% of the total errors and involves inaccuracies in interpreting visual data. This type of error can be reduced through the implementation of more advanced vision encoders. Furthermore, the fact that Calculation Error constitutes only 4.3% of the errors suggests that MathGLM-Vision-32B demonstrates considerable robustness in numerical and computational tasks. Lastly, Question Misunderstood Error, which constitutes 2.5% of the total, occurs when the model fails to correctly interpret question. Enhancing natural language processing capabilities and refining context understanding can significantly reduce these types of errors. Addressing these identified error types through targeted enhancements can significantly boost the overall effectiveness of MathGLM-Vision-32B. Figure 6 demonstrate some cases of the Calculation Error category. More detailed examples of these errors can be found in Appendix G.

## 5 RELATED WORKS

**Multi-Modal Language Model.** The development of Multi-Modal Language Models (MLLMs) have emerged as a significant area of research, which are designed to integrate information from multiple modalities—typically text and images—to perform tasks that require a holistic understanding of both visual and linguistic inputs. Pioneering efforts such as ViLBERT Lu et al. (2019) and LXMERT Tan & Bansal (2019) have advanced this field by conducting the joint pre-training on image-text datasets. They process text and image inputs separately before fusing them for final task layers, significantly improving performance on tasks like image captioning and visual question answering. The continues evolution of MLLMs has lead to innovations in data fusion techniques. Notable models such as CLIP Radford et al. (2021), ALIGN Jia et al. (2021), and BLIP Li et al. (2022) have adopted contrastive learning paradigms to align visual and language information from billions of image-text pairs. Concurrently, the success of LLMs Brown et al. (2020); Du et al. (2021); Zeng et al. (2022); Le Scao et al. (2023); Bai et al. (2022); Touvron et al. (2023a); Ouyang et al. (2022);

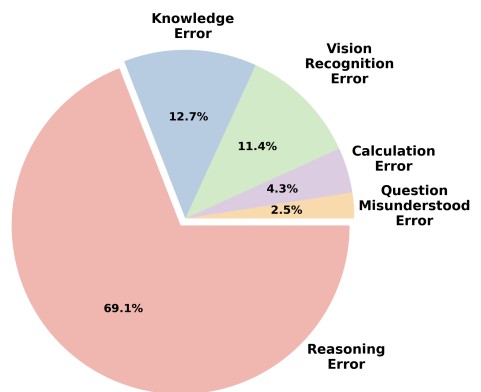

Figure 5: Error distribution of MathGLM-Vision-32B.

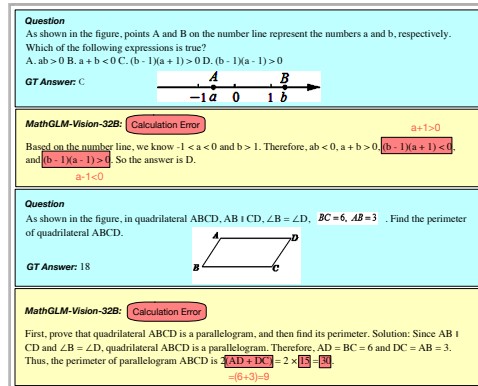

Figure 6: Cases of Calculation Error category.

Hoffmann et al. (2022); Smith et al. (2022); Chowdhery et al. (2023) facilitates the integration of LLMs into multi-modal tasks by utilizing pre-training alignment and visual instruction tuning, leading to the emergence of multi-modal language models (MLLMs) Liu et al. (2024b); Liu et al.; Wang et al. (2023c); Li et al. (2023); Dai et al. (2024); Bai et al. (2023a). Despite MLLMs have demonstrated remarkable capabilities on tasks such as image caption and visual question answering, they stall face significant challenges in solving mathematical problems that involve visual information Yue et al. (2024a); Lu et al. (2023); Zhang et al. (2024); Wang et al. (2024).

**Mathematical Reasoning.**   Recently, math-specific LLMs Azerbayev et al. (2023); Wang et al. (2023a); Yue et al. (2024b); Ying et al. (2024); Yu et al. (2023); Yue et al. (2023); Yuan et al. (2023); Luo et al. (2023) have demonstrated remarkable abilities in handling mathematical reasoning tasks that only involve textual information. These models have been specifically trained on web-scale instruction mathematical dataset or fine-tuned on specialized mathematical problem sets. For instance, WizardMath Luo et al. (2023) and MetaMath Yu et al. (2023) have implemented data augmentation methods to enhance the models' ability to understand and solve mathematical problems by enriching the MATH Hendrycks et al. (2021) and GSM8K Cobbe et al. (2021) datasets. Recent research has also focused on creating specialized MLLMs for mathematical tasks. UniGeo Chen et al. (2022) and UniMath Liang et al. (2023) have demonstrated enhanced datasets and conventional deep learning approaches for geometric problem solving. MLLMs like G-LLaVA Gao et al. (2023a), GeoGPT4V Cai et al. (2024), and Math-LLaVA Shi et al. (2024) are tailored for mathematical problem solving, incorporating both geometric understanding and algebraic reasoning. Additionally, several benchmark datasets Yue et al. (2024a); Lu et al. (2023); Zhang et al. (2024); Wang et al. (2024) are proposed to evaluate the multi-modal mathematical reasoning abilities of MLLMs.

## 6 CONCLUSION

In this paper, we attempt to address the issues in current mathematical MLLMs. We construct a fine-tuning dataset named MathVL, upon which we conduct a Supervised Fine-Tuning (SFT) process. This initiative results in the development of a series of enhanced MLLMs, designated as MathGLM-Vision. Specially, MathGLM-Vision contains three variations: MathGLM-Vision-9B, MathGLM-Vision-19B, and MathGLM-Vision-32B, each fine-tuned on different backbone models: GLM-4-V, CogVLM2, and CogVLM-32B, respectively. These developed MathGLM-Vision significantly improve the capabilities of mathematical reasoning, achieving substantial performance improvements. Relative to their respective backbone models, MathGLM-Vision-9B, MathGLM-Vision-19B, and MathGLM-Vision-32B show improvements of 39%, 65%, and 53.7% on the Geometry Problem Solving (GPS) minitest split of MathVista, demonstrating the effectiveness of MathVL in enhancing the mathematical problem-solving abilities of MLLMs. Additionally, we evaluate the effectiveness of MathGLM-Vision on our curated MathVL-test benchmark. Experimental results reveal that MathGLM-Vision not only surpass their backbone models in specialized mathematical tests but also preserve the generalizability capabilities in general vision-language understanding domains.

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

## A    DATASET CASES

In this section, we provide a detailed overview of specific cases from our constructed MathVL dataset. These cases demonstrate the variety of mathematical disciplines covered by MathVL, including arithmetic, geometry, algebra, statistics, and word problems. Figure 7, Figure 8, Figure 9, Figure 10, and Figure 11 depict the types of problems that MathGLM-Vision is designed to tackle in each respective category. Each of these categories is critical for assessing the comprehensive mathematical capabilities of MathGLM-Vision. By tackling a wide range of problems, MathGLM-Vision demonstrates its versatility and robustness in addressing diverse mathematical tasks.

## B    BACKBONE MODELS

We utilize the following multi-modal large language models as our backbone models for conducting Specialized Fine-Tuning (SFT) on the constructed MathVL. The detailed description of each backbone model can be represented as follows:

- **GLM-4V-9B** is a bilingual (Chinese and English) multi-modal large language model, developed collaboratively by Zhipu.AI and Tsinghua University. It is built upon the foundational architecture of GLM-4-9B, enhancing its capabilities to handle complex multi-modal interactions. GLM-4V-9B takes a high resolution of 1120 * 1120 images as visual inputs. In comprehensive evaluations that test various capabilities including combined language skills, perceptual reasoning, text recognition, and chart understanding, GLM-4V-9B consistently outperforms competitors such as GPT-4-turbo-2024-04-09, Gemini 1.0 Pro, Qwen-VL-Max, and Claude 3 Opus, demonstrating its superior performance across multiple modalities.

- **CogVLM2** is a series of open-source multi-modal large language models derived from Meta-Llama-3-8B-Instruct, developed by Zhipu.AI and Tsinghua University. This series contains two models: cogvlm-llama3-chat-19B and cogvlm2-llama3-chinese-chat-19B. The former is a monolingual language model focused on English and the latter is a bilingual model supporting both English and Chinese. CogVLM2 is designed to handle extended content lengths up to 8K and accepts high-resolution images up to 1344 * 1344. Here, we choose cogvlm2-llama3-chinese-chat-19B as our backbone to pre-train our MathGLM-Vision-19B.

- **CogVLM-32B** is a close-source multi-modal large language model, developed by Zhipu.AI and Tsinghua University. It is based on the GLM-32B architecture and is optimized for handling complex multi-modal tasks. CogVLM-32B is engineered to process visual inputs at a high resolution of 1120 * 1120, enabling detailed image analysis and enhanced interaction with visual data.

Table 8 demonstrates an overview of the MathGLM-Vision series, detailing the different model parameters and configurations.

| Model | LLM Size | ToTal Size | Language Model | | | Image Encoder | | |
|---|---|---|---|---|---|---|---|---|
| | | | Layers | Hidden Size | Heads | Layers | Hidden Size | Heads |
| MathGLM-Vision-9B | 9B | 20B | 40 | 4096 | 32 | 63 | 1792 | 16 |
| MathGLM-Vision-19B | 8B | 19B | 32 | 4096 | 32 | 63 | 1792 | 16 |
| MathGLM-Vision-32B | 32B | 43B | 58 | 6656 | 52 | 63 | 1792 | 16 |

Table 8: An overview of MathGLM-Vision series along with model parameters and configurations.

## C    DESCRIPTIONS OF VQA DATASETS

Here, we provide a detailed description of collected visual question answering datasets (VQA) datasets. Table 9 demonstrates details 19 different VQA datasets, including task types and visual context.

**Question**

Given that a, b, and c are points on the number line as shown in the figure, simplify |a - b| - 2|c - b| + |b - c| to get _______.

**Answer**

[Solution]Given that from the figure, b < a < 0 < c, we have a - b > 0 and b - c < 0. Therefore, the original expression simplifies to a - b - 2(c - b) - b + c = a - c.

[Answer]The answer is a - c.

**Question**

According to the program flow shown in the figure, if the initial input value is x = 4, the final output result is ( )
A. 60   B. 1540   C. 1500   D. 2020

**Answer**

[Solution] Substitute x = 4 into the program flow: (4 × (4 + 1)) / 2 = 10, which is less than 100. Substitute x = 10 into the program flow: (10 × (10 + 1)) / 2 = 55, which is less than 100. Substitute x = 55 into the program flow: (55 × (55 + 1)) / 2 = 1540, which is greater than 100. Therefore, the final output result is 1540.

[Answer]The answer is B.

**Question**

As shown in the figure, in the long division of 28 by 16, the step indicated by the arrow represents ( )
A. 80 ones   B. 80 tenths   C. 80 hundredths

**Answer**

[Solution]The 8 indicated by the arrow is in the tenths place, and after dropping the 0, it combines with 8 to form 80 hundredths; based on this, the solution can be given. In the long division of 28 by 16, the step indicated by the arrow represents 80 hundredths.

[Answer]The answer is C.

**Question**

Given that the positions of numbers a and b on the number line are as shown in the figure, then a + b is ( )
A. Negative   B. Positive   C. 0   D. Indeterminate.

**Answer**

[Solution]Given that b < 0 < a and |a| < |b|, therefore, a + b < 0. Thus, a + b is a negative number. So the answer is A. From the given conditions, we know b < 0 < a and |a| < |b|, which leads to the conclusion. This problem tests the understanding of the number line. Using the number line to determine the relationship between a and b is key to solving the problem.

[Answer] A

**Question**

As shown in the figure, below is Yutan's test paper. The score is ( )
A. 20 points   B. 80 points   C. 0 points   D. 40 points

**Answer**

[Solution]The reciprocal of -3 is -1/3, so the first question is incorrect. The arithmetic square root of √9 is 3, so the second question is incorrect. Therefore, Yutan's score is 0 points. Thus, the answer is C. First, according to the rule of finding the reciprocal of an integer, which is one over that integer, the reciprocal of -3 is -1/3. Then, according to the definition and method of finding the arithmetic square root, the arithmetic square root of √9 is 3. This problem mainly tests the understanding and method of finding the arithmetic square root and the reciprocal, which should be mastered well.

[Answer]C.

Figure 7: Cases of arithmetical problems in our MathVL dataset.

**Question**

As shown in the figure, it is known that BC is parallel to GE, and ∠AFG = ∠1 = 50°.
(1) Prove that AF is parallel to DE;
(2) If AQ bisects ∠FAC, intersects BC at point Q, and ∠Q = 15°, find the measure of ∠ACQ.

**Answer**

[Solution](1) According to the properties of parallel lines, we get ∠E = ∠1 = 50°. Given that ∠AFG = 50°, we have ∠AFG = ∠E. Finally, according to the criteria for parallel lines, we can draw the conclusion. (2) According to the properties of the exterior angle of a triangle, we get ∠AHD = 65°. Using the properties of parallel lines and angle bisectors, we find ∠CAQ = 65°. By using the sum of the interior angles of a triangle, we can get the conclusion. This question tests the determination and properties of parallel lines, the definition of angle bisectors, and the properties of the exterior angle of a triangle. Mastery of the determination and properties of parallel lines is key to solving the problem.

[Answer] (1) Proof: ∵ BC ∥ GE, ∴ ∠E = ∠1 = 50°, ∵ ∠AFG = ∠1 = 50°, ∴ ∠E = ∠AFG = 50°, ∴ AF ∥ DE; (2) Solution: ∵ ∠1 = 50°, ∠Q = 15°, ∴ ∠AHD = 65°, ∵ AF ∥ DE, ∴ ∠FAQ = ∠AHD = 65°, ∵ AQ bisects ∠FAC, ∴ ∠CAQ = ∠FAQ = 65°, ∴ ∠ACQ = 180° - ∠CAQ - ∠Q = 180° - 65° - 15° = 100°.

**Question**

As shown in the figure, in the equilateral triangle ABC, angle ∠C is cut along the dashed line DE. Then, ∠ADE + ∠DEB = ______ .

**Answer**

[Solution] Given that △ABC is an equilateral triangle, ∠A = ∠B = 60°. Therefore, ∠ADE + ∠BED = 360° - 60° × 2 = 240°. Thus, the answer is 240°. By first noting that ∠A and ∠B are each 60° due to the properties of an equilateral triangle, and then using the fact that the sum of the interior angles of a quadrilateral is 360°, subtracting the measures of ∠A and ∠B gives the result. This problem mainly tests the understanding of the interior angles of polygons and the properties of equilateral triangles, with the key point being that the sum of the interior angles of a quadrilateral is 360°.

[Answer] 240°.

**Question**

As shown in the figure, the large rectangle is composed of 2 identical large squares, 2 identical small squares, and 5 identical small rectangles. If the perimeter of this large rectangle is 48 cm, and the sum of the areas of the four squares is 68 cm², find the area of one of the small rectangles.

**Answer**

[Solution] Let the length of the small rectangle be a cm and the width be b cm. According to the problem, we have a + b = 8 and $a^2 + b^2 = $ 34. Therefore, ab = (1/2)[(a + b)² - (a² + b²)] = 15.
[Answer] The area of the small rectangle is 15 cm².

**Question**

As shown in the figure, the quadrilateral paper piece ABCD is folded along MN, and points A and D fall on A₁ and D₁ respectively. If ∠1 + ∠2 = 145°, then ∠B + ∠C = ______°.

**Answer**

[Solution]Given that ∠1 + ∠2 = 145°, we have ∠AMN + ∠DNM = (360° - 145°) / 2 = 107.5°. Since ∠A + ∠D + (∠AMN + ∠DNM) = 360° and ∠A + ∠D + (∠B + ∠C) = 360°, it follows that ∠B + ∠C = ∠AMN + ∠DNM = 107.5°.
[Answer]The answer is 107.5°.

**Question**

The three views of a geometric body are shown in the figure. The surface area of the geometric body is ( )
A. 24 + 3π/4   B. 24 + 5π/4   C. 24 + π   D. 8 + 3π/4

**Answer**

[Solution]Based on the three views of the geometric body, the geometric body is composed of a cube and an eighth of a sphere. Therefore, the surface area is S = 6 × 4 + (1/8) × 4π × 1² + (1/4) × π × 1² = 24 + 3π/4.
[Answer] The answer is A.

Figure 8: Cases of geometric problems in our MathVL dataset.

***Question***

In the same Cartesian coordinate system, the graphs of the linear function $y = k_1x + b$ and the proportional function $y = k_2x$ are shown in the figure. The range of x values that satisfy $k_1x + b > k_2x$ is ______.

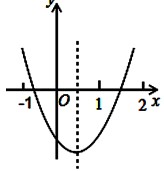

***Answer***

[Solution]When $x \leq -3$, the line $l_1$: $y_1 = k_1x + b$ is above the line $l_2$: $y_2 = k_2x$, meaning $k_1x + b > k_2x$. Therefore, the range of x values that satisfy $k_1x + b > k_2x$ is $x < -3$.
[Answer]The answer is $x < -3$.

***Question***

As shown in the figure, the graph represents the quadratic function $y = ax^2 + bx + c$. Among the following statements: ① $ac > 0$; ② $a - b + c < 0$; ③ $4ac < b^2$; ④ $2a + b > 0$; ⑤ When $x > 0$, y decreases as x increases. The number of correct statements is ( )
A. 1    B. 2    C. 3    D. 4

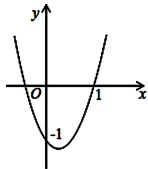

***Answer***

[Solution]① From the graph, we know that $a > 0$ and $c < 0$, so $ac < 0$. Thus, ① is incorrect. ② From the graph, we know that when $x = -1$, $y = a - b + c > 0$, so ② is incorrect. ③ Since the parabola has two intersection points with the x-axis, $\triangle = b^2 - 4ac > 0$. Thus, ③ is correct. ④ From the axis of symmetry, we know that $-b/2a < 1$, so $2a + b > 0$. Thus, ④ is correct. ⑤ When $x > -b/2a$, y increases as x increases, so ⑤ is incorrect.
[Answer]The answer is B.

***Question***

The graph of the quadratic function $y = ax^2 + bx + c$ is shown in the figure. Among the following conclusions: ① $ab > 0$; ② $a + b - 1 = 0$; ③ $a > 1$; ④ One root of the quadratic equation $ax^2 + bx + c = 0$ is 1, and the other root is $-1/a$. The correct conclusions are ______.

***Answer***

[Solution]① From the graph of the quadratic function, we know that the parabola opens upwards, so $a > 0$. The axis of symmetry is to the right of the y-axis, so $b < 0$. Therefore, $ab < 0$, making ① incorrect. ② From the graph, we know that the parabola intersects the x-axis at $(1, 0)$ and the y-axis at $(0, -1)$. Therefore, $c = -1$ and $a + b - 1 = 0$, making ② correct. ③ Since $a + b - 1 = 0$, we have $a - 1 = -b$. Given that $b < 0$, we have $a - 1 > 0$, so $a > 1$, making ③ correct. ④ Since the parabola intersects the y-axis at $(0, -1)$, the equation of the parabola is $y = ax^2 + bx - 1$. Since the parabola intersects the x-axis at $(1, 0)$, one root of $ax^2 + bx - 1 = 0$ is 1. According to the relationship between the roots and coefficients, the other root is $-1/a$, making ④ correct.
[Answer]The correct conclusions are ②, ③, and ④.

***Question***

As shown in the figure, the solution set of the inequality $2x - a < -1$ is represented. The value of a is ( )
A. $a \leq -1$    B. $a \leq -2$    C. $a = -1$    D. $a = -2$

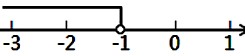

***Answer***

[Solution]Given $2x - a < -1$, we have $2x < a - 1$, which means $x < (a - 1)/2$. From the number line, we know $x < -1$. Therefore, $(a - 1)/2 = -1$. Solving this, we get $a = -1$.
[Answer]The answer is C.

***Question***

The store purchased a batch of T-shirts at a unit price of 20 yuan. After a trial sale, it was found that the daily sales volume y (units) and the sales price x (yuan/unit) satisfy the linear function relationship shown in the figure. (1) Find the function relationship between y and x (no need to write the range of x values). (2) Without considering factors like inventory, at what sales price will the daily profit W be maximized?

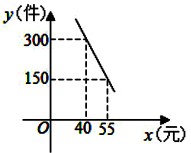

***Answer***

[Solution](1) Let the function relationship between y and x be $y = kx + b$ ($k \neq 0$). Substituting $(40, 300)$ and $(55, 150)$ into the equation, we get: $300 = 40k + b$ and $150 = 55k + b$. Solving, we get: $k = -10$ and $b = 700$. Therefore, the function relationship between y and x is $y = -10x + 700$. (2) From the problem, we get: $W = (x - 20) \cdot y = (x - 20)(-10x + 700) = -10x^2 + 900x - 14000 = -10(x - 45)^2 + 6250$. Since $-10 < 0$, when $x = 45$, W reaches its maximum value of 6250 yuan.
[Answer]Without considering factors like inventory, the daily profit W is maximized when the sales price is set at 45 yuan.

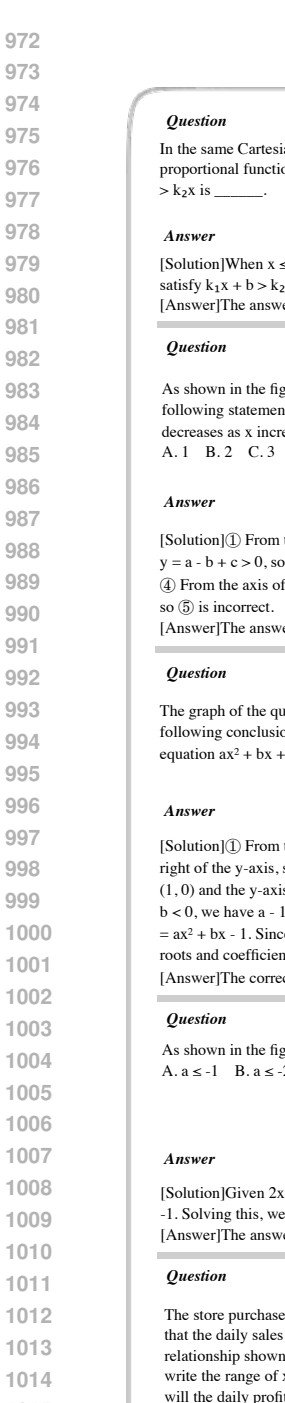

Figure 9: Cases of algebraic problems in our MathVL dataset.

**Question**

A city plans to build a flower bed in the central square, dividing it into 6 parts (as shown in the figure). Four different colors of flowers need to be planted, one color per part, and adjacent parts cannot have the same color. The number of different planting methods is __________. (Answer with a number)

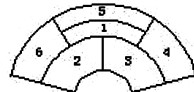

**Answer**

[Solution]From the problem's context, we need to plant flowers of 4 different colors in 6 parts. From the figure, we can see that there must be 2 pairs of parts with the same color. We can classify and calculate the planting methods based on the parts with the same color. (1) If parts ② and ⑤ are the same color, then parts ③ and ⑥ or parts ④ and ⑥ must also be the same color. Therefore, the number of methods for this case is $N_1 = 4 \times 3 \times 2 \times 2 \times 1 = 48$. (2) If parts ③ and ⑤ are the same color, then parts ② and ④ or parts ⑥ and ④ must also be the same color. Therefore, the number of methods for this case is $N_2 = 4 \times 3 \times 2 \times 2 \times 1 = 48$. (3) If parts ② and ④ and parts ③ and ⑥ are the same color, then the number of methods for this case is $N_3 = 4 \times 3 \times 2 \times 1 = 24$. Therefore, the total number of planting methods is $N = N_1 + N_2 + N_3 = 48 + 48 + 24 = 120$.

[Answer]The answer is 120.

**Question**

In a shooting competition, Person A and Person B each shoot 5 times. The bar chart of their scores is shown in the figure. The number of correct statements among the following four is ( ) ① The mean of Person A's scores is equal to the mean of Person B's scores. ② The median of Person A's scores is greater than the median of Person B's scores. ③ The variance of Person A's scores is less than the variance of Person B's scores. ④ The range of Person A's scores is equal to the range of Person B's scores.

A. 1   B. 2   C. 3   D. 4

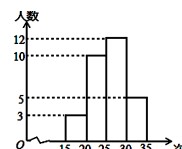

**Answer**

[Solution]In statement ①, the mean score for Person B is $(5 + 5 + 5 + 6 + 9) / 5 = 6$, and the mean score for Person A is $(4 + 5 + 6 + 7 + 8) / 5 = 6$. Therefore, the mean score for Person A is equal to the mean score for Person B, so ① is correct. In statement ②, the median score for Person A is 6, and the median score for Person B is 5. Therefore, the median score for Person A is greater than the median score for Person B, so ② is correct. In statement ③, the variance for Person A's scores is $(2^2 \times 2 + 1^2 \times 2) / 5 = 2$, and the variance for Person B's scores is $(1^2 \times 3 + 3^2 \times 1) / 5 = 2.4$. Therefore, the variance for Person A's scores is less than the variance for Person B's scores, so ③ is correct. In statement ④, the range for Person A's scores is 4, and the range for Person B's scores is 4. Therefore, the range for Person A's scores is equal to the range for Person B's scores, so ④ is correct.

[Answer]The answer is D.

**Question**

To understand the physical fitness of ninth-grade students at a certain school, a random sample of ninth-grade students was tested for the number of sit-ups they could do in one minute. The results were displayed in the histogram shown. Based on the chart, calculate the percentage of students whose number of sit-ups falls within the range of 25 to 30. The percentage is ( )

A. 40%   B. 30%   C. 20%   D. 10%

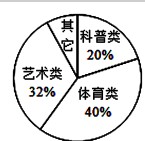

**Answer**

[Solution]From the frequency histogram, we can determine that the total number of students surveyed is $3 + 10 + 12 + 5 = 30$. The number of students who did 25 to 30 sit-ups is 12, so the percentage is 40%.

[Answer]A.

**Question**

As shown in the pie chart, the number of seventh-grade (Class 1) students participating in extracurricular activities at a certain school is represented. If the number of students participating in art activities is 16, then the number of students participating in sports activities is _______.

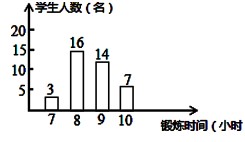

**Answer**

[Solution]Given that the number of seventh-grade (Class 1) students participating in art activities is 16, which represents 32% of the total number of students in the class, the total number of students in seventh-grade (Class 1) is $16 \div 32\% = 50$. Since the percentage of students participating in sports activities is 40%, the number of students participating in sports activities is $50 \times 40\% = 20$.

[Answer]The answer is 20.

**Question**

As shown in the chart regarding the weekly physical exercise of students in a certain class, the mode and median of the time spent on physical exercise by the students this week are ( )

A. 8, 9   B. 8, 8.5   C. 16, 8.5   D. 16, 14

**Answer**

[Solution]The mode is the number that appears most frequently in a data set, which is 8. When the data set is arranged in ascending order, the number in the middle position is the median. According to the definition of median, the median of this data set is 9.

[Answer]The answer is A.

Figure 10: Cases of statistical problems in our MathVL dataset.

**Question**

During the May Day holiday, there was an appliance promotion, and mom bought two appliances as shown in the figure. (1) How much did mom spend in total? (2) How much cheaper was the microwave oven compared to its original price? (3) Can you propose and solve other math problems based on this scenario?

原价：5900 元 原价：2200 元
现价：4000 元 现价：1900 元

**Answer**

(1) 5900 yuan (2) 300 yuan (3) Question: How much cheaper was the washing machine compared to its original price? Answer: 1900 yuan

---

**Question**

As shown in the figure, the operating hours of Haichang Polar Ocean Park are (________) hours.

营业时间
10：00——下午 7：00
中午休息 1 小时

**Answer**

This question requires calculating the operating hours, which is the elapsed time. Using the formula elapsed time = end time - start time, convert 7 PM to 24-hour time as 19:00. Thus, 19:00 - 10:00 = 9 hours, then subtract a 1-hour lunch break. Therefore, the answer is 8 hours.

---

**Question**

The store is preparing to bag 120 apples for sale (as shown in the figure). The apples can be evenly divided into bags of (______) apples each; using bags of (______) apples each will leave (______) apples remaining.

8个/袋 9个/袋

**Answer**

Divide 120 by 8 and 9 respectively to get the answer. $120 \div 8 = 15$ (bags) $120 \div 9 = 13$ (bags) with a remainder of 3 (apples) Therefore, 120 apples can be evenly divided into bags of 8 apples each, and when using bags of 9 apples each, there will be 3 apples remaining. So the answer is: 8, 9, 3.

---

**Question**

As shown in the figure, at the 15th Asian Games, China won ( ) more gold medals than the combined total of gold medals won by South Korea and Japan.
A. 108  B. 57  C. 115  D. 107

13—15 届亚运会中、韩、日三国金牌统计图

**Answer**

[Solution] $165 - (58 + 50) = 165 - 108 = 57$ (medals) Thus, China won 57 more gold medals than the combined total of South Korea and Japan. [Answer] B

---

**Question**

Mr. Zhang and Mr. Ma took a taxi together from the school. Mr. Zhang went to Xinhua Bookstore, and Mr. Ma went to Yintai Department Store (as shown in the figure). They agreed to share the taxi fare reasonably. The taxi fare rates are: 0-3 kilometers (base fare) 10 yuan, and beyond 3 kilometers, 1.8 yuan per kilometer (rounded up to the nearest kilometer). Please help them calculate how much each person should pay.

学校  新华书店    银泰百货
  ├─5千米─┤
  ├──────15千米──────┤

**Answer**

[Solution] $10 + (5 - 3) \times 1.8 = 13.6$ yuan $1.8 \times (15 - 5) = 18$ yuan $13.6 \div 2 = 6.8$ yuan $6.8 + 18 = 24.8$ yuan. Thus, Mr. Zhang should pay 6.8 yuan, and Mr. Ma should pay 24.8 yuan. [Answer] 6.8 yuan; 24.8 yuan.

Figure 11: Cases of word problems in our MathVL dataset.

| Dataset | Task | Visual Context |
|---------|------|----------------|
| DocVQA | Figure Question Answering (FQA) | Document Image |
| DVQA | Figure Question Answering (FQA) | Bar Chart |
| FigureQA | Figure Question Answering (FQA) | Charts and Plots |
| PlotQA | Figure Question Answering (FQA) | Bar, Line, Scatter |
| MapQA | Figure Question Answering (FQA) | Map Chart |
| IconQA | Math Word Problem (MWP) | Abstract Scene |
| TabMWP | Math Word Problem (MWP) | Table |
| CLEVR-Math | Math Word Problem (MWP) | Synthetic Scene |
| TQA | Textbook Question Answering (TQA) | Scientific Figure |
| AI2D | Textbook Question Answering (TQA) | Scientific Figure |
| ScienceQA | Textbook Question Answering (TQA) | Scientific Figure |
| A-OKVQA | Visual Question Answering (VQA) | Natural Image |
| VQA2.0 | Visual Question Answering (VQA) | Natural Image |
| PMC-VQA | Visual Question Answering (VQA) | Medical Image |
| VizWiz | Visual Question Answering (VQA) | Natural Image |
| Super-CLEVR | Visual Question Answering (VQA) | Synthetic Scene |
| VQA-AS | Visual Question Answering (VQA) | Abstract Scene |
| VQA-RAD | Visual Question Answering (VQA) | Medical Image |
| TextVQA | Visual Question Answering (VQA) | Natural Image |

Table 9: Summary of VQA datasets.

# D    IMPLEMENTATION DETAILS

We provide a detailed overview of the Specialized Fine-Tuning (SFT) process applied to our MathGLM-Vision. The specific hyperparameters used during this process are outlined in Table 10.

| parameters | MathGLM-Vision-9B | MathGLM-Vision-19B | MathGLM-Vision-32B |
|------------|-------------------|--------------------|--------------------|
| Total steps | 35,000 | 35,000 | 35,000 |
| Global Batch Size | 128 | 128 | 128 |
| Learning Rate | $1e^{-5}$ | $1e^{-5}$ | $1e^{-5}$ |
| Learning Rate Schedule | cosine decay | cosine decay | cosine decay |
| Warmup Ratio | 0.01 | 0.01 | 0.01 |
| Weight Decay | $5e^{-2}$ | $5e^{-2}$ | $5e^{-2}$ |
| Optimizer | AdamW | AdamW | AdamW |
| Input Resolution | 1120 * 1120 | 1344 * 1344 | 1120 * 1120 |
| Image Length | 1600 | 2304 | 1600 |

Table 10: The detailed setup of the SFT procedures.

# E    THE DETAILED DESCRIPTION OF BENCHMARK DATASETS

In this section, we provide an in-depth description of the benchmark datasets used to evaluate the performance of MathGLM-Vision. These benchmark datasets have been carefully curated to test the MLLMs' capabilities. The detailed description of benchmark datasets is provides as follows.

- **MathVista**
  MathVista is a comprehensive benchmark dataset designed to rigorously evaluate the mathmetical reasoning capabilities of language models (LMs), especially in varied visual contexts. This dataset offers a comprehensive evaluation benchmark designed to integrate mathematical reasoning with visual understanding, focusing on five primary tasks: figure question

answering (FQA); geometry problem solving (GPS); math word problem (MWP); textbook question answering (TQA); and visual question answering (VQA).

- **MathVista-GPS**

  MathVista-GPS, a subset of the MathVista Dataset, specifically focuses on the domain of geometry problem solving. The questions in this subset range from basic shape recognition to more advanced problems involving theorems, calculations and reasoning.

- **MathVerse**

  MathVerse is designed to provide a fair and comprehensive assessment of MLLMs' capabilities in visual mathematics. The benchmark comprises 2,612 high-quality, multi-subject math problems, each featuring diagrams and converted into six different versions by human annotators. These versions offer varying levels of multi-modal information, allowing for a thorough evaluation of MLLMs' understanding of visual diagrams.

- **MATH-Vision**

  The MATH-Vision (MATH-V) dataset comprises 3,040 high-quality mathematical problems, each featuring a visual context and sourced from 19 real math competitions. This extensive and diverse collection allows for a comprehensive evaluation of LMMs' ability to interpret and reason with visual information in mathematical contexts.

- **MMMU**

  The Massive Multi-discipline Multi-modal Understanding and Reasoning (MMMU) benchmark encompasses 11.5K questions across six disciplines, including Art, Business, Health & Medicine, Science, Humanities & Social Science, and Tech & Engineering. The tasks in MMMU challenge models to perform sophisticated multi-modal analysis and apply domain-specific knowledge, demanding a higher level capability in comprehension and integration.

# F DETAILED EXPERIMENTAL RESULTS ON PUBLIC BENCHMARK DATASETS

**Results on the testmini subset of MathVista.** To comprehensively evaluate the performance of MathGLM-Vision across various task types featured in the MathVista dataset, we systematically evaluate it on the testmini subset. This subset has been carefully selected to represent a diverse range of mathematical problem types, ensuring a robust assessment of our model's capabilities. Table 11 shows the evaluation results on the testmini subset of MathVista across various task types. Notably, MathGLM-Vision-19B and MathGLM-Vision-32B surpass human performance in overall accuracy, highlighting the advanced capabilities of these models in handling complex mathematical problems. In particular, MathGLM-Vision excels significantly in geometry problem solving (GPS) and geometry reasoning (GEO), demonstrating its superior proficiency in mathematical reasoning.

| Model | Input | ALL | FQA | GPS | MWP | TQA | VQA | ALG | ARI | GEO | LOG | NUM | SCI | STA |
|---|---|---|---|---|---|---|---|---|---|---|---|---|---|---|
| Human Performance | $Q, I$ | 60.30 | 59.70 | 48.40 | 73.00 | 63.20 | 55.90 | 50.90 | 59.20 | 51.40 | 40.70 | 53.80 | 64.90 | 63.90 |
| 2-shot CoT GPT-4 | $Q, I_c, I_t$ | 30.50 | 27.21 | 35.91 | 21.30 | 43.13 | 28.17 | 35.72 | 25.17 | 35.80 | 24.74 | 15.41 | 47.28 | 31.29 |
| 2-shot PoT GPT-4 | $Q, I_c, I_t$ | 31.74 | 27.58 | 37.35 | 23.87 | 43.00 | 30.27 | 37.15 | 27.93 | 37.48 | 22.68 | 15.83 | 44.47 | 31.87 |
| GPT-4V | $Q, I$ | 49.90 | 43.10 | 50.50 | 57.50 | 65.20 | 38.00 | 53.00 | 49.00 | 51.00 | 21.60 | 20.10 | 63.10 | 55.80 |
| LLaVA-LLaMA-2-13B | $Q, I$ | 25.40 | 22.86 | 24.57 | 18.15 | 35.82 | 29.69 | 26.93 | 22.47 | 24.45 | 19.07 | 19.05 | 34.71 | 21.61 |
| **MathGLM-Vision-9B** | $Q, I$ | 52.20 | 46.10 | 64.42 | 58.60 | 55.70 | 37.43 | 59.79 | 43.91 | 62.34 | 10.81 | 37.50 | 54.10 | 54.82 |
| **MathGLM-Vision-19B** | $Q, I$ | 61.10 | 59.85 | **65.38** | 68.28 | 53.80 | **55.31** | 59.79 | 59.21 | **63.18** | 18.92 | **59.03** | 53.28 | 68.44 |
| **MathGLM-Vision-32B** | $Q, I$ | **62.40** | **62.83** | 62.02 | **69.35** | **62.03** | 54.19 | **60.50** | **60.62** | 61.92 | 16.22 | 52.08 | **60.66** | **72.09** |

Table 11: **Accuracy scores on the _testmini_ subset of MathVista.** Input: $Q$: question, $I$: image, $I_c$: image caption, $I_t$: OCR texts detected from the image. ALL: overall accuracy. Task types: FQA: figure question answering, GPS: geometry problem solving, MWP: math word problem, TQA: textbook question answering, VQA: visual question answering. Mathematical reasoning types: ALG: algebraic reasoning, ARI: arithmetic reasoning, GEO: geometry reasoning, LOG: logical reasoning, NUM: numeric common sense, SCI: scientific reasoning, STA: statistical reasoning. The highest accuracy among all baseline MLLMs is marked in red, while the highest accuracy among various variants of MathGLM-Vision is marked bold.

**Results on the testmini set of MathVerse.** To thoroughly evaluate the performance of MathGLM-Vision across 12 detailed subjects within the MathVerse dataset, we conduct comprehensive experiments and report the results in Table 12. This analysis delves into the model's ability to address a broad spectrum of mathematical challenges, ranging from geometry to functions. As shown in Table 12, MathGLM-Vision surpasses all open-source MLLMs and most close-source MLLMs. However, it still falls short by 14% compared to the performance of GPT-4V. In some subjects such as Angle, Analytic, and Property, MathGLM-Vision achieves better performance compared the advanced GPT-4V. For example, MathGLM-Vision-32B shows remarkable performance in plane geometry, particularly in handling angle-related problems, where it achieves a 60.1% accuracy, showcasing its strong geometric reasoning capabilities.

| Model | All | Plane Geometry | | | | | | Solid Geometry | | | | Functions | | | | |
|---|---|---|---|---|---|---|---|---|---|---|---|---|---|---|---|---|
| | | All | Len | Area | Angle | Anal | Apply | All | Len | Area | Vol | All | Coord | Prop | Exp | Apply |
| *Closed-source MLLMs* | | | | | | | | | | | | | | | | |
| Qwen-VL-Plus | 21.3 | 17.3 | 19.1 | 16.4 | 16.1 | 23.6 | 13.2 | 24.8 | 18.1 | 18.7 | 33.4 | 31.3 | 52.5 | 25.1 | 10.8 | 50.3 |
| Gemini-Pro | 35.3 | 33.0 | 32.2 | 42.6 | 28.4 | 30.2 | 32.3 | 33.4 | 35.0 | 29.3 | 36.1 | 28.3 | 25.7 | 26.6 | 10.8 | 51.3 |
| Qwen-VL-Max | 37.2 | 38.4 | 41.7 | 46.4 | 32.6 | 40.6 | 38.7 | 33.7 | 25.4 | 28.3 | 42.6 | 38.4 | 43.7 | 35.5 | 13.6 | 61.0 |
| GPT-4V | 54.4 | 56.9 | 60.8 | 63.4 | 52.6 | 48.5 | 60.9 | 50.2 | 54.8 | 39.9 | 56.8 | 52.8 | 72.3 | 47.1 | 30.9 | 70.1 |
| *Open-source MLLMs* | | | | | | | | | | | | | | | | |
| LLaMA-Adapter V2 | 5.8 | 5.9 | 4.0 | 5.9 | 6.6 | 13.4 | 3.3 | 4.6 | 5.3 | 3.1 | 5.7 | 6.2 | 6.7 | 6.1 | 4.5 | 7.9 |
| ImageBind-LLM | 10.0 | 9.7 | 12.1 | 9.9 | 9.2 | 10.2 | 4.8 | 4.6 | 4.9 | 3.5 | 5.3 | 14.9 | 12.3 | 13.8 | 4.6 | 25.9 |
| mPLUG-Owl2 | 10.3 | 7.7 | 8.2 | 6.0 | 5.7 | 12.4 | 10.6 | 11.0 | 9.2 | 6.7 | 15.7 | 17.4 | 22.8 | 18.6 | 5.3 | 22.2 |
| MiniGPT-v2 | 10.9 | 11.6 | 10.0 | 9.8 | 14.3 | 9.1 | 11.8 | 1.7 | 2.2 | 1.6 | 0.5 | 11.2 | 4.2 | 15.7 | 4.0 | 21.1 |
| LLaVA-1.5 | 12.7 | 11.8 | 13.1 | 15.1 | 9.7 | 9.4 | 13.2 | 10.6 | 12.1 | 8.7 | 11.6 | 14.8 | 12.7 | 9.5 | | 23.7 |
| SPHINX-Plus | 14.0 | 14.4 | 14.2 | 10.5 | 14.1 | 16.5 | 16.8 | 7.0 | 7.2 | 6.1 | 7.6 | 17.9 | 11.1 | 19.1 | 6.3 | 27.7 |
| G-LLaVA | 15.7 | 20.2 | 17.3 | 13.6 | 26.5 | 5.9 | 23.1 | 5.0 | 10.3 | 4.4 | 3.1 | 9.2 | 9.1 | 9.1 | 1.3 | 15.5 |
| LLaVA-NeXT | 17.2 | 15.9 | 14.8 | 13.1 | 16.3 | 17.7 | 17.8 | 19.6 | 33.3 | 11.7 | 12.6 | 23.1 | 24.5 | 23.4 | 8.0 | 33.1 |
| ShareGPT4V | 17.4 | 16.9 | 16.2 | 17.9 | 16.9 | 12.2 | 21.1 | 15.0 | 13.6 | 10.9 | 19.7 | 20.2 | 19.9 | 22.2 | 8.4 | 25.8 |
| SPHINX-MoE | 22.8 | 24.5 | 26.3 | 28.4 | 21.1 | 26.6 | 24.4 | 15.8 | 9.4 | 10.7 | 26.3 | 19.5 | 23.5 | 19.3 | 9.2 | 30.3 |
| InternLM-XC2 | 25.9 | 26.2 | 27.1 | 29.7 | 20.6 | 18.5 | 22.2 | 20.1 | 34.5 | 14.1 | 25.2 | 23.7 | 24.4 | 24.9 | 10.6 | 36.3 |
| *MathGLM-Vision* | | | | | | | | | | | | | | | | |
| **MathGLM-Vision-9B** | **44.2** | 45.3 | **43.7** | 48.9 | 41.5 | **53.5** | 52.2 | 42.0 | 54.2 | 50.0 | 29.4 | 42.1 | 25.0 | 42.3 | **43.8** | 47.5 |
| **MathGLM-Vision-19B** | 42.5 | 41.8 | 34.8 | **55.3** | 38.9 | 46.5 | **53.6** | 51.3 | **66.7** | **52.3** | **43.1** | 38.4 | 18.8 | 38.0 | 28.1 | 55.0 |
| **MathGLM-Vision-32B** | **49.2** | **49.0** | 42.4 | 59.6 | **51.3** | 48.8 | 50.7 | 45.4 | 62.5 | 40.9 | 41.2 | **52.8** | **43.8** | **59.2** | 40.6 | **55.0** |

Table 12: **Mathematical Evaluation on Different Subjects and Subfields in MathVerse's** *testmini* **Set.** Len: Length; Anal: Analytic; Apply: Applied; Vol: Volume; Coord: Coordinate; Prop: Property; Exp: Expressio. The highest accuracy among all baseline MLLMs is marked in red, while the highest accuracy among various variants of MathGLM-Vision is marked bold.

**Results on Math-Vision datasets.** To effectively assess MathGLM-Vision's ability across diverse subjects and difficulty levels within the Math-Vision dataset, we conduct a series of detailed evaluation experiments and report results in Table 13. Specifically, GPT-4V leads the close-source models with an overall accuracy of 22.76%, yet it remains significantly below the human performance benchmark of 75.66%. MathGLM-Vision shows competitive performance across a variety of mathematical disciplines compared to most of close-source MLLMs, with MathGLM-Vision-32B achieving the overall accuracy of 26.5%, closely approaching that of GPT-4V. Notably, MathGLM-Vision-32B excels in solid geometry with a accuracy of 29.1%, significantly outperforming the accuracy of 23.8% on GPT-4V. This superior performance in solid geometry highlights MathGLM-Vision-32B's advanced spatial reasoning and geometric processing capabilities, which are essential for tackling complex three-dimensional problems.

# G ERROR CASES

Figure 12, Figure 13, Figure 14, and Figure 15 show examples of errors made by MathGLM-Vision-32B on the MathVL-test dataset. Each figure highlights a specific type of error, providing valuable insights into the model's limitations and areas for improvement.

# H CASE STUDY

Figure 16, Figure 17, Figure 18, and Figure 19 present several case studies from MathGLM-Vision-32B. These figures showcase the model's performance in various scenarios, highlighting its strengths

**Question**

As shown in the figure, when a weight is placed in a measuring cup and water is added, the relationship between the height of the water in the cup (y) and the volume of water added (x) is approximately ( ).

A.    B.    C.    D.

**GT Answer:** B

**MathGLM-Vision-32B:** Reasoning Error

Analysis: According to the problem, the relationship between the height of the water in the cup (y) and the volume of water added (x) should be divided into two segments: initially slow, then fast. Solution: Since a weight is placed in the cup first, the height of the water (y) initially increases slowly with the volume of water added (x). Only when the water surface surpasses the weight does the height increase rapidly with the addition of more water. Therefore, the answer is C.

Figure 12: **An example of reasoning error.** MathGLM-Vision-32B mistakenly believed that the height of the water would initially increase rapidly and then slowly, indicating a reasoning error.

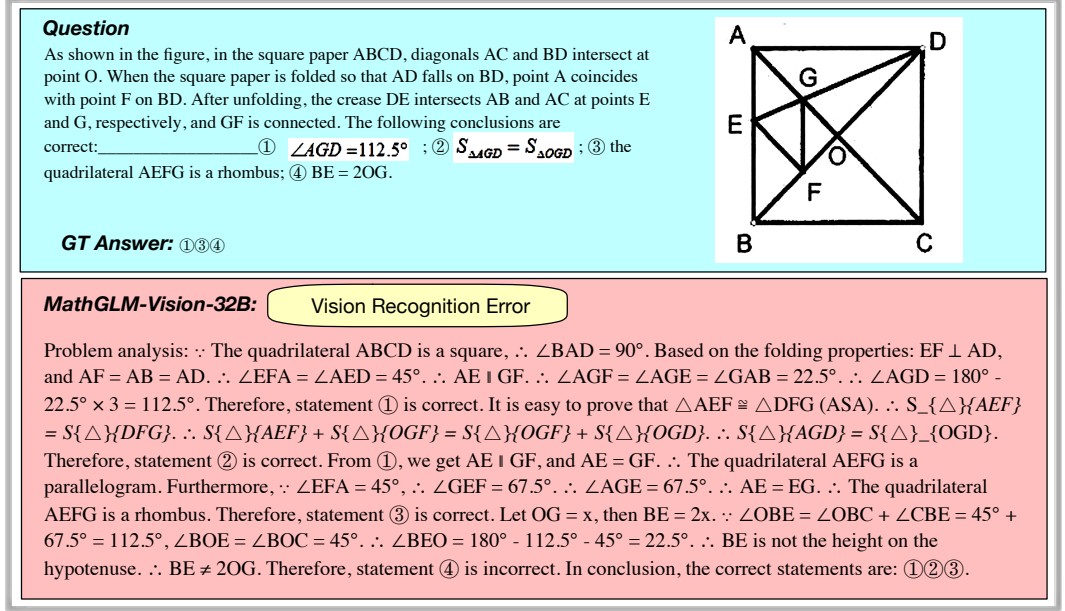

**Question**

As shown in the figure, in the square paper ABCD, diagonals AC and BD intersect at point O. When the square paper is folded so that AD falls on BD, point A coincides with point F on BD. After unfolding, the crease DE intersects AB and AC at points E and G, respectively, and GF is connected. The following conclusions are correct:__________ ① $\angle AGD = 112.5°$ ; ② $S_{\triangle AGD} = S_{\triangle OGD}$ ; ③ the quadrilateral AEFG is a rhombus; ④ BE = 2OG.

**GT Answer:** ①③④

**MathGLM-Vision-32B:** Vision Recognition Error

Problem analysis: ∵ The quadrilateral ABCD is a square, ∴ ∠BAD = 90°. Based on the folding properties: EF ⊥ AD, and AF = AB = AD. ∴ ∠EFA = ∠AED = 45°. ∴ AE ∥ GF. ∴ ∠AGF = ∠AGE = ∠GAB = 22.5°. ∴ ∠AGD = 180° - 22.5° × 3 = 112.5°. Therefore, statement ① is correct. It is easy to prove that △AEF ≅ △DFG (ASA). ∴ $S_{\triangle AEF}$ = $S_{\triangle DFG}$. ∴ $S_{\triangle AEF} + S_{\triangle OGF} = S_{\triangle OGF} + S_{\triangle OGD}$. ∴ $S_{\triangle AGD} = S_{\triangle OGD}$. Therefore, statement ② is correct. From ①, we get AE ∥ GF, and AE = GF. ∴ The quadrilateral AEFG is a parallelogram. Furthermore, ∵ ∠EFA = 45°, ∴ ∠GEF = 67.5°. ∴ ∠AGE = 67.5°. ∴ AE = EG. ∴ The quadrilateral AEFG is a rhombus. Therefore, statement ③ is correct. Let OG = x, then BE = 2x. ∵ ∠OBE = ∠OBC + ∠CBE = 45° + 67.5° = 112.5°, ∠BOE = ∠BOC = 45°. ∴ ∠BEO = 180° - 112.5° - 45° = 22.5°. ∴ BE is not the height on the hypotenuse. ∴ BE ≠ 2OG. Therefore, statement ④ is incorrect. In conclusion, the correct statements are: ①②③.

Figure 13: **An example of vision recognition error.** MathGLM-Vision-32B incorrectly interpreted the geometric properties of the diagram, leading to a vision recognition error.

**Question**

As shown in Figure 1, in the isosceles trapezoid ABCD, AB ∥ CD, AD = BC, and DC = 2. A moving point P starts from point C and moves along CB and BA until it stops at point A. Let the distance traveled by point P be x, and the area of △DCP be y. If the graph of the function y with respect to x is shown in Figure 2, then the area of the isosceles trapezoid ABCD is ( ).

A. 12   B. 18   C. 20   D. 21

**GT Answer:** C

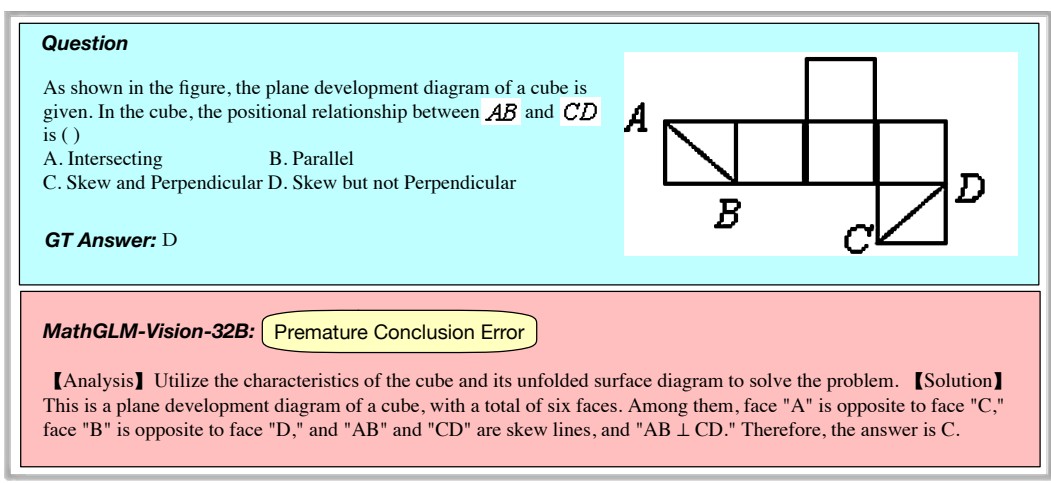

**MathGLM-Vision-32B:** [ Calculation Error ]

【Analysis】 According to the graph of the function and the figure, we find the length of AB. Using the properties of the isosceles trapezoid, we find that DC = AD = BC = 2. Using the Pythagorean theorem, we find AC, and thus the answer.

【Solution】 From the graph, we see that when point P moves 5 units, the area of △DCP starts to remain unchanged, so we deduce that BC = 5. As point P moves from C to B, y increases with x, and when x = 5, the area changes, so AB = 5. ∵ The quadrilateral ABCD is an isosceles trapezoid, ∴ ∠B = ∠BAD. Also, ∵ ∠B + ∠BAP = ∠BAD + ∠CDA, ∴ ∠CDA = ∠BAP. Also, ∵ ∠ADC = ∠BAP, ∴ △DCA ∼ △PBA. Thus, CD : AB = DA : PA, which means 2 : 5 = (5 - 2) : PA. Solving for PA, we get PA = 3.75. ∴ The area of the isosceles trapezoid ABCD is the sum of the areas of △ABC and △DAB, which is 1/2 × (2 + 5) × 3.75 + 1/2 × 3.75 × 2 = 18.75 + 3.75 = 22.5. Therefore, the answer is D.

Figure 14: **An example of calculation error.** MathGLM-Vision-32B made an error in calculating the lengths and areas in the isosceles trapezoid, leading to an incorrect solution.

**Question**

As shown in the figure, the plane development diagram of a cube is given. In the cube, the positional relationship between $AB$ and $CD$ is ( )

A. Intersecting                B. Parallel
C. Skew and Perpendicular D. Skew but not Perpendicular

**GT Answer:** D

**MathGLM-Vision-32B:** [ Premature Conclusion Error ]

【Analysis】 Utilize the characteristics of the cube and its unfolded surface diagram to solve the problem. 【Solution】 This is a plane development diagram of a cube, with a total of six faces. Among them, face "A" is opposite to face "C," face "B" is opposite to face "D," and "AB" and "CD" are skew lines, and "AB ⊥ CD." Therefore, the answer is C.

Figure 15: **An example of premature conclusion error.** MathGLM-Vision-32B prematurely concluded that AB is perpendicular to CD without proper reasoning, leading to a premature conclusion error.

| Model | Overall | Alg | AnaG | Ari | CombG | Comb | Cnt | DescG | GrphT | Log | Angle | Area | Len | SolG | Stat | Topo | TransG |
|---|---|---|---|---|---|---|---|---|---|---|---|---|---|---|---|---|---|
| Human Performance | | | | | | | | | | | | | | | | | |
| Human (testmini) | 75.66 | 57.9 | 79.0 | 100.0 | 100.0 | 47.4 | 94.7 | 89.5 | 63.2 | 63.2 | 36.8 | 52.6 | 73.7 | 89.5 | 89.5 | 100.0 | 73.7 |
| Open-source MLLMs | | | | | | | | | | | | | | | | | |
| LLaVA-v1.5-7B | 8.52 | 7.0 | 7.1 | 10.7 | 7.1 | 4.8 | 10.5 | 7.7 | 10.0 | 9.2 | 15.6 | 10.2 | 9.8 | 5.3 | 8.6 | 4.4 | 4.8 |
| SPHINX (V2) | 9.70 | 6.7 | 7.1 | 12.9 | 7.5 | 7.7 | 6.0 | 9.6 | 16.7 | 10.1 | 11.0 | 11.8 | 12.5 | 8.2 | 8.6 | 8.7 | 6.0 |
| ShareGPT4V-7B | 10.53 | 5.5 | 3.6 | 12.9 | 10.1 | 4.8 | 7.5 | 11.5 | 14.4 | 10.9 | 16.2 | 11.8 | 12.3 | 9.8 | 15.5 | 17.4 | 11.3 |
| LLaVA-v1.5-13B | 11.12 | 7.0 | 14.3 | 14.3 | 9.1 | 6.6 | 6.0 | 13.5 | 5.6 | 13.5 | 10.4 | 12.6 | 14.7 | 11.5 | 13.8 | 13.0 | 10.7 |
| ShareGPT4V-13B | 11.88 | 7.5 | 15.5 | 16.4 | 10.7 | 8.9 | 9.0 | 11.5 | 8.9 | 7.6 | 11.6 | 13.0 | 17.4 | 10.3 | 8.6 | 8.7 | 12.5 |
| SPHINX-MoE | 14.18 | 7.8 | 17.9 | 14.3 | 15.6 | 9.5 | 11.9 | 12.5 | 15.6 | 12.6 | 16.2 | 15.6 | 17.8 | 13.5 | 12.1 | 8.7 | 16.1 |
| InternLM-XComposer2-VL | 14.54 | 9.3 | 15.5 | 12.1 | 15.3 | 11.3 | 10.5 | 14.4 | 22.2 | 19.3 | 19.7 | 15.6 | 15.0 | 11.9 | 15.5 | 26.1 | 15.5 |
| Closed-source MLLMs | | | | | | | | | | | | | | | | | |
| Qwen-VL-Plus | 10.72 | 11.3 | 17.9 | 14.3 | 12.7 | 4.8 | 10.5 | 15.4 | 8.9 | 14.3 | 11.6 | 6.4 | 10.0 | 14.3 | 6.9 | 8.7 | 11.31 |
| Qwen-VL-Max | 15.59 | 10.7 | 19.1 | 20.0 | 16.9 | 12.5 | 17.9 | 16.4 | 12.2 | 21.0 | 13.3 | 14.2 | 19.8 | 11.5 | 20.7 | 13.0 | 17.3 |
| Gemini Pro | 17.66 | 15.1 | 10.7 | 20.7 | 20.1 | 11.9 | 7.5 | 20.2 | 21.1 | 16.8 | 19.1 | 19.0 | 20.0 | 14.3 | 13.8 | 17.4 | 20.8 |
| GPT4V | 22.76 | 27.3 | 32.1 | 35.7 | 21.1 | 16.7 | 13.4 | 22.1 | 14.4 | 16.8 | 22.0 | 22.2 | 20.9 | 23.8 | 24.1 | 21.7 | 25.6 |
| MathGLM-Vision | | | | | | | | | | | | | | | | | |
| **MathGLM-Vision-9B** | 19.2 | 18.6 | 20.2 | 19.3 | 15.3 | 18.5 | 20.9 | 26.0 | 18.9 | 15.1 | 23.1 | 20.4 | 18.3 | 23.8 | 19.0 | 17.4 | 14.3 |
| **MathGLM-Vision-19B** | 21.6 | 22.0 | **29.8** | 23.6 | 22.4 | 18.5 | **25.4** | 25.0 | 17.8 | 16.0 | 20.2 | 22.0 | 20.3 | 21.3 | 20.7 | **30.4** | 23.2 |
| **MathGLM-Vision-32B** | **26.5** | **22.9** | 20.2 | **24.3** | **23.1** | **28.0** | 20.9 | **34.6** | **27.8** | **23.5** | **31.2** | **26.8** | **30.1** | **29.1** | **22.4** | 17.4 | **26.2** |

Table 13: **Comparison of model performances across various mathematical subjects.** Subjects: Alg: algebra, AnaG: analytic geometry, Ari: arithmetic, CombG: combinatorial geometry, Comb: combinatorics, Cnt: counting, DescG: descriptive geometry, GrphT: graph theory, Log: logic, Angle: metric geometry - angle, Area: metric geometry - area, Len: metric geometry - length, SolG: solid geometry, Stat: statistics, Topo: topology, TransG: transformation geometry. The highest accuracy among all baseline MLLMs is marked in red, while the highest accuracy among various variants of MathGLM-Vision is marked bold.

in providing concise and clear answers with logical mathematical reasoning. Compared to other close-source MLLMs, MathGLM-Vision-32B stands out for its ability to deliver precise and understandable solutions.

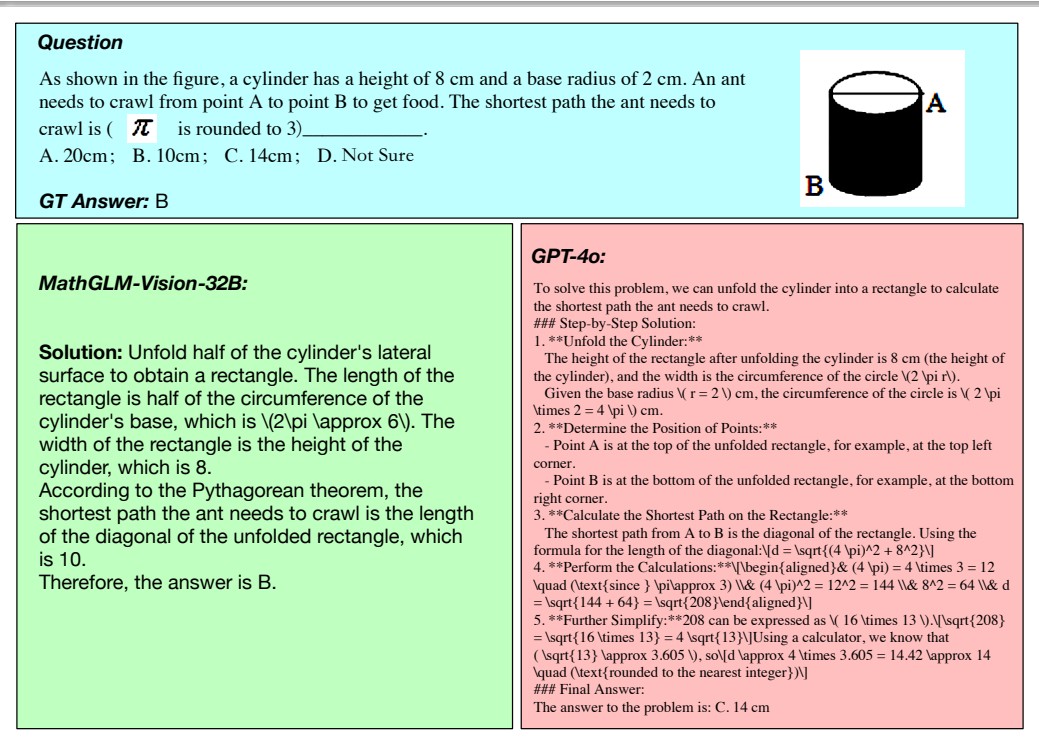

Figure 16: **An example of solid geometry problem.** MathGLM-Vision-32B correctly calculated the shortest path with shorter steps, while GPT-4o made a reasoning error, leading to a different conclusion.

**Question**

In the $4 \times 4$ square grid shown in the image, $\angle 1 + \angle 2 + \angle 3 + \angle 4 + \angle 5 + \angle 6 + \angle 7 =$
A. 330°   B. 315°   C. 310°   D. 320°

**GT Answer:** B

**MathGLM-Vision-32B:**

【Analysis】 Based on the axial symmetry of the square, we have $\angle 1 + \angle 7 = 90°$, $\angle 2 + \angle 6 = 90°$, $\angle 3 + \angle 5 = 90°$, and $\angle 4 = 45°$. 【Solution】 Since the triangle containing $\angle 1$ is congruent to the triangle containing $\angle 7$, $\angle 1 + \angle 7 = 90°$. Similarly, $\angle 2 + \angle 6 = 90°$, and $\angle 3 + \angle 5 = 90°$. Additionally, since $\angle 4 = 45°$, we have $\angle 1 + \angle 2 + \angle 3 + \angle 4 + \angle 5 + \angle 6 + \angle 7 = 315°$. Therefore, the answer is B.

**Claude-3.5-Sonnet:**

Let's solve this problem step by step: Step 1: Understand the problem The problem presents a 4x4 square grid with 7 rays emanating from the bottom left corner. We need to calculate the sum of these 7 angles. Step 2: Analyze the angle of each ray - Ray 1: 45° (1 square up, 1 square right) - Ray 2: 63.43° (1 square up, 2 squares right, arctan(1/2)) - Ray 3: 71.57° (1 square up, 3 squares right, arctan(1/3)) - Ray 4: 75.96° (1 square up, 4 squares right, arctan(1/4)) - Ray 5: 51.34° (2 squares up, 3 squares right, arctan(2/3)) - Ray 6: 59.04° (2 squares up, 4 squares right, arctan(2/4)) - Ray 7: 36.87° (3 squares up, 4 squares right, arctan(3/4)) Step 3: Calculate the sum of the angles 45° + 63.43° + 71.57° + 75.96° + 51.34° + 59.04° + 36.87° = 403.21° Step 4: Adjust the result Since we are calculating the angles between these rays, we need to subtract 360° (a full circle) from 403.21°. 403.21° - 360° = 43.21° Step 5: Choose the closest answer The closest option to 43.21° is 45°, which corresponds to 315° (because 360° - 45° = 315°). The answer to this problem is: B. 315°.

Figure 17: **An example of a planar geometry problem.** MathGLM-Vision-32B correctly utilized axial symmetry to determine the angle sum, while Claude-3.5-Sonnet arrived at the correct answer through an erroneous calculation process. Despite the correct final answer, the calculation process was incorrect and overly complex.

1512
1513
1514
1515
1516
1517
1518
1519
1520
1521
1522
1523
1524
1525
1526
1527
1528
1529
1530
1531
1532
1533
1534
1535
1536

**Question**

As shown in the figure, in the parallelogram ABCD, E is a point on CD, and AE and BD are connected, intersecting at point F. If $DE : EC = 2 : 3$, then $S_{\triangle DEF} : S_{\triangle ABF}$ = __________.

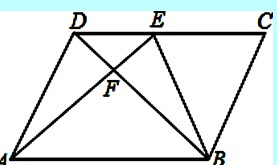

**GT Answer:** 4:25

**MathGLM-Vision-32B:**

【Solution】\n Analysis: From the properties of the parallelogram, we can deduce that △DFE ~ △BFA. Based on the fact that the area ratio of similar triangles is equal to the square of the ratio of their corresponding sides, we can find the answer.\n ∵ The quadrilateral ABCD is a parallelogram,\n ∴ DC ∥ AB, and CD = AB.\n ∴ △DFE ~ △BFA,\n ∴ $S_{\triangle DEF} : S_{\triangle ABF} = (DE : AB)^2 = 4 : 25$.\n Therefore, the answer is 4 : 25.

**GPT-4o:**

To solve the plane geometry problem, we proceed as follows: **Step 1: Define the known quantities** In the parallelogram ABCD: - E is a point on segment CD - AE and BD intersect at point F - The ratio of DE to EC is 2:3, i.e., DE:EC = 2:3 **Step 2: Set the unknown quantities** Let DE = 2x, EC = 3x, then CD = DE + EC = 5x. **Step 3: Calculate the area ratio** To calculate \( \frac{S_{DEF}}{S_{ABF}} \), we first determine the position of the intersection point F of AE and BD. According to the problem, point F is at the common vertex of triangles DEF and ABF, and we can use the area ratio of similar triangles to solve it. **Step 4: Analyze similar triangles** Since DE:EC = 2:3, and triangle AEF is similar to triangle CDF, the area ratio is proportional to the square of the segments: - The area ratio of triangles DEF and CDF \( \frac{S_{DEF}}{S_{CDF}} = \left(\frac{DE}{CD}\right)^2 = \left(\frac{2}{5}\right)^2 = \frac{4}{25} \) - The area ratio of triangles CDF and ABC \( \frac{S_{CDF}}{S_{ABC}} = \left(\frac{CD}{AB}\right)^2 = \left(\frac{1}{1}\right)^2 = 1 \) - Since the base sides of the two triangles are equal, \( \frac{S_{CDEF}}{S_{CABC}} = 4/20 = 1/5 \) **Step 5: Comprehensive calculation** Therefore, the area ratio of triangles DEF and ABF is: - \( \frac{S_{DEF}}{S_{ABF}} = \frac{S_{DEF}}{S_{ABF}} = \frac{2 \times 4 \times 1}{5 \times 20 \times 5} = 2/15 \). **Step 6: Determine the final ratio** In summary, the answer to this problem is \( \frac{S_{DEF}}{S_{ABF}} = 2/15 = 2:15 \) Therefore, the answer is 2:15.

Figure 18: **An example of a planar geometry problem.** MathGLM-Vision-32B correctly used the properties of similar triangles to find the area ratio, while GPT-4o misinterpreted the geometric relationships and misapplied the formula for the area ratio, leading to an erroneous conclusion.

1537
1538
1539
1540
1541
1542
1543
1544
1545
1546
1547
1548
1549
1550
1551
1552
1553
1554
1555
1556
1557
1558
1559
1560
1561
1562
1563
1564
1565

**Question**

As shown in the figure, a right triangle ABC (where ∠ABC = 60° and ∠C = 90°) is rotated clockwise around point B by a certain angle to position $A_1BC_1$, making points A, B, and $C_1$ collinear. This angle is ( )
A. 30° B. 60° C. 90° D. 120°

**GT Answer:** D

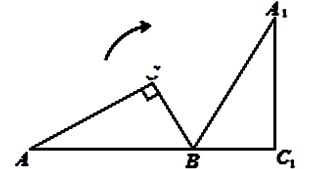

**MathGLM-Vision-32B:**

【Analysis】Use the properties of rotation to calculate. 【Solution】In the triangle, ∠ABC = 60°, and the rotation angle is ∠$CBC_1$. Therefore, ∠$CBC_1$ = 180° - 60° = 120°. This rotation angle is equal to 120°. Therefore, the answer is D.

**QWen-VL-Max:**

This is a problem about geometric transformation. We need to find the angle of rotation for triangle ABC. Step 1: Determine the initial position In the initial position, angle ABC in triangle ABC is 60 degrees, and angle C is 90 degrees. Step 2: Understand the problem The problem requires us to rotate triangle ABC clockwise around point B so that points A, B, and C1 lie on the same straight line. Step 3: Calculate the rotation angle Since we want points A, B, and C1 to be collinear, it means that angle AB_1C1 must be 180 degrees. Since angle ABC is 60 degrees, angle BAC is 30 degrees. Therefore, angle AB_1C1 = 180 - (angle BAC + angle ABC) = 180 - (30 + 60) = 180 - 90 = 90 degrees. Step 4: Draw the conclusion Based on the above calculations, we can conclude that the rotation angle should be 90 degrees. The answer to this problem is C. 90°.

Figure 19: **An example of a geometric transformation problem.** MathGLM-Vision-32B correctly used the properties of rotation to determine the angle, while QWen-VL-Max misunderstood the problem requirements and incorrectly calculated the rotation angle as 90°.

# I  MODEL EVALUATION

**Evaluation on public benchmarks**  The existing public benchmarks for evaluating a wide array of open-source and close-source models are neither timely nor comprehensive enough. To compare our MathGLM-Vision with the state-of-the-art open-source and close-source LLMs, we have supplemented the evaluations for some models missing from the public benchmark leaderboard.

We generate LLMs' responses through API access (for closed-source models) and local inference (for open-source models). The evaluation was then conducted following the official evaluation code from each benchmark's GitHub repository. The source of the models used in the evaluation can be found in Table 14.

| Model | Input | LLM Size | Source |
|---|---|---|---|
| *Closed Source Models* | | | |
| *Multi-modal LLMs* | | | |
| Gemini Pro | $Q, I$ | - | gemini-pro |
| Gemini 1.5 Pro | $Q, I$ | - | gemini-1.5-pro |
| GPT-4V | $Q, I$ | - | gpt-4-vision-preview |
| GPT-4-turbo | $Q, I$ | - | gpt-4-turbo |
| GPT-4o | $Q, I$ | - | gpt-4o |
| Claude-3-Opus | $Q, I$ | - | claude-3-opus-20240229 |
| Claude-3.5-Sonnet | $Q, I$ | - | claude-3-5-sonnet-2024620 |
| Qwen-VL-Plus | $Q, I$ | - | qwen-vl-plus |
| Qwen-VL-Max | $Q, I$ | - | qwen-vl-max |
| *Open Source Models* | | | |
| *General Multi-modal LLMs* | | | |
| mPLUG-Owl | $Q, I$ | 7B | mPLUG-Owl |
| LLaMA-Adapter-V2 | $Q, I$ | 7B | LLaMA-Adapter V2 |
| InstructBLIP | $Q, I$ | 7B | InstructBLIP |
| LLaVA-1.5 | $Q, I$ | 13B | LLaVA-v1.5-13B |
| ShareGPT-4V | $Q, I$ | 13B | ShareGPT4V-13B |
| SPHINX-MoE | $Q, I$ | 8*7B | SPHINX-MoE |
| SPHINX-Plus | $Q, I$ | 13B | SPHINX-Plus |
| InternLM-XC2 | $Q, I$ | 7B | InternLM-XComposer2-VL-7B |
| InternVL-1.2-Plus | $Q, I$ | 34B | InternVL-Chat-V1-2-Plus |
| *Geo-Multi-modal LLMs* | | | |
| G-LLaVA | $Q, I$ | 7B | G-LLaVA-7B |
| G-LLaVA | $Q, I$ | 13B | G-LLaVA-13B |
| LLaVA-1.5-G | $Q, I$ | 7B | LLaVA-1.5-7B-GeoGPT4V |
| LLaVA-1.5-G | $Q, I$ | 13B | LLaVA-1.5-13B-GeoGPT4V |
| ShareGPT4V-G | $Q, I$ | 7B | ShareGPT4V-7B-GeoGPT4V |
| ShareGPT4V-G | $Q, I$ | 13B | ShareGPT4V-1.5-13B-GeoGPT4V |
| Math-LLaVA | $Q, I$ | 13B | Math-LLaVA-13B |

Table 14: The source of the models used in the evaluation.

**Evaluation on MathVL-test** We evaluate MathGLM-Vision and several close-source MLLMs using our specially constructed MathVL-test. The evaluation process of our MathVL-test is conducted through 3 key-step: generation, extraction, and scoring.

For the generation step, the model responses are generated by providing the model with queries which incorporate the Chain of Thought (CoT) template, questions, and diagram information. The reponses of close-source MLLMs is generated through API access. For the extraction step, we use GPT-3.5-turbo to extract the model's answer based on the reponses of first step. Finally, in the scoring step, the score for each question is determined by GLM-4 based on the comparison between the extracted answer and the standard answer.

The prompts used to guide the LLM in response generation, answer extraction and scoring can be found in Table 15.

| Task | Prompt |
|---|---|
| Response Generation | You are a very skilled math teacher. Please provide a detailed, step-by-step solution to the question, following a step-by-step format. Be sure to conclude with a summary that states "The answer to this question is" followed by the final result. |
| Answer Extraction | Please read the following example. Then extract the answer from the model response and type it at the end of the prompt. Hint: Please answer the question requiring an integer answer and provide the final value, e.g., 1, 2, 3, at the end. Question: Which number is missing? Model response: The number missing in the sequence is 14. Extracted answer: 14 Hint: Please answer the question requiring a floating-point number with one decimal place and provide the final value, e.g., 1.2, 1.3, 1.4, at the end. Question: What is the fraction of females facing the camera? Model response: The fraction of females facing the camera is 0.6, which means that six out of ten females in the group are facing the camera. Extracted answer: 0.6 |
| Scoring | Please determine if the extracted_answer correctly answers the question. The correct answer needs to be extracted from the answer without re-calculating it, and the answer in the answer should be considered the final answer. Also, do not judge whether the answer is correct. The question may contain multiple sub-questions, and correctly answering the question includes correctly answering every sub-question and every result within each sub-question. A relative error divided by the absolute value of the original answer of less than 0.01 is allowed. If the prediction does not contain an answer, it is considered wrong. If the answer is not numerical, determine the equivalence of the expression, not just the value. If there is one mistake, the answer is wrong. Only if all results given in the prediction are correct is it considered correct. There is no need to consider whether the solution process of the prediction is complete. Please first extract the answers given by the prediction, determine the relative error, check if each sub-question is answered correctly, and finally give the judgment in a single line (output only "yes" or "no" in a single line). |

Table 15: Prompts for response generation, answer extraction and scoring.

