# OpenReview forum: "MathGLM-Vision: Solving Mathematical Problems with Multi-Modal Large Language Model"
_ICLR.cc/2025/Conference — Submitted to ICLR 2025_

### Official Review · Reviewer_JCVo · 2024-11-02

**Soundness:** 3
**Presentation:** 2
**Contribution:** 2
**Rating:** 5
**Confidence:** 5

**Summary:**

This paper introduces MathVL,  a diverse and comprehensive multi-modal mathematical dataset for SFT. MLLMs of different scales trained on MathVL have demonstrated excellent performance, proving the dataset's effectiveness for solving mathematical problems. Beside, the paper also establishes a benchmark dataset called MathVL-test to evaluate the mathematical reasoning abilities of MLLMs using a multi-image format.

**Strengths:**

1. The proposed model, MathGLM-Vision, achieves good results on several benchmarks. The performance on MathVL-test is promising.
2. This paper presents multiple cases to better understand the collected dataset.

**Weaknesses:**

1. The writing of the paper requires improvement. Several citations are not formatted correctly (e.g., Lines 75-78).
2. The paper appears to be more engineering-focused, with the primary contribution being the introduction of the MathVL dataset. While this is a valuable resource, the abstract does not mention any plans to make the dataset open-source or publicly available. Without access to the dataset, the practical impact of the work is significantly diminished, as other researchers cannot replicate or build upon the proposed model. Therefore, making the dataset publicly available would greatly strengthen the paper’s contribution and its value to the broader research community.
3. Maybe more figure examples can help to understand better the challenges presented in Line82-85. For example, the paper claims that current MLLMs tend to overlook the diversity of visual information in mathematics. Some visualizations or analyses would better support this conclusion.

**Questions:**

My primary concern revolves around the paper's contribution to the community. While I acknowledge that the paper introduces a model for addressing mathematical problems and offers a useful benchmark for the field, I believe that the most impactful contribution to the advancement of Math MLLMs would be the provision of the dataset for others to use in training their own models. Making the data publicly available would significantly enhance the reproducibility and accessibility of the research, fostering broader progress in the development of Math MLLMs.

---

### Official Review · Reviewer_PSD6 · 2024-11-02

**Soundness:** 3
**Presentation:** 3
**Contribution:** 2
**Rating:** 5
**Confidence:** 4

**Summary:**

Current MLLMs specialized in math focus too much on geometry and miss other kinds of visual math information. To improve this, this paper created a new dataset called MathVL and developed MathGLM-Vision, which performed better on math problems than other models.

**Strengths:**

1. This article provides the MLLM community with high-quality Math VL data. The introduced dataset, MathVL, is very useful and significantly enhances the mathematical abilities of some MLLMs.
2. The experiments in this article are very detailed, and the writing is excellent.

**Weaknesses:**

The paper claims three contributions: dataset, model, and benchmark. However, the dataset was only validated on weaker models, and the differences from existing datasets are not clearly described (details below). The developed model is essentially a fine-tuning version of the proposed dataset without any architectural innovation (while SFT is a common practice, it does not constitute a contribution since the model is merely a product of the dataset, especially as the final model is not state-of-the-art). Therefore, the claimed contribution are not justified well.

The difference between this article and Math-LLaVA is mentioned in lines 81-88, where it states that Math-LLaVA focuses on solving geometric problems. However, after reading Math-LLaVA, I found that it also includes various tasks, such as Math Word Problems and Textbook Question Answering.


1. For Table 4: 1) I am particularly interested in the effectiveness of MathVL on other stronger models, such as the 7B InternLM-XC2, since the base model used in the experiment seems to have low mathematical ability according to the results in the table (about 40% accuracy on MathVista). 2) Why is the accuracy for GLM-4V-9B at 46.7 on MathVista, with its fine-tuned MathGLM-Vision-9B version reaching 52.2, while CogVLM2, which has an accuracy of only 40.85, achieves 61.1 after fine-tuning as MathGLM-Vision-19B? Why does a weaker model show better performance after fine-tuning?
2. Table 5, results for GLM-4V-9B, GLM-4V-9B w/o VQA datasets, and MathGLM-Vision-9B are missing.
3. Table 6, it seems to lack the exact composition method of MathVL-test in the paper. I am curious whether it is sampled from Open-source Data or Open-source Data + Chinese K12 Data. If it is the latter, then the performance drop with only SFT on Open-source Data alone would not necessarily indicate the importance of Chinese K12 Data, given that a significant performance drop is only observed on MathVL-test without Chinese K12 Data.

**Questions:**

1. The VQA dataset's size (i.e., the number of data points it contains) would be helpful to clarify;
2. For the error analysis, could you confirm which tool or method was used to identify error types? Was it done through a specific automated tool, manual annotation, or perhaps a LLM?

---

### Official Review · Reviewer_496G · 2024-11-03

**Soundness:** 3
**Presentation:** 3
**Contribution:** 3
**Rating:** 6
**Confidence:** 4

**Summary:**

The paper presents MathGLM-Vision, a specialized multi-modal large language model designed to solve mathematical problems. Specifically, the authors introduce MathVL, a fine-tuning dataset with a diverse range of mathematical problems involving visual elements. MathGLM-Vision models are fine-tuned on MathVL and are evaluated across public benchmarks. Results demonstrate significant performance improvements in mathematical problem-solving.

**Strengths:**

1.	MathVL is a contribution as it diversifies the visual components beyond traditional geometry, covering topics like arithmetic, algebra, and statistics. This expansion enhances the MLLM’s utility across a broader spectrum of math problems.
2.	The experimental results show that MathGLM-Vision, with visual input, consistently outperforms its text-only counterparts, validating the value of multi-modal inputs.
3.	The paper uses a comprehensive set of benchmarks, including MathVL-test and public datasets, to evaluate performance. This approach provides a well-rounded assessment of MathGLM-Vision's mathematical reasoning capabilities and general vision-language understanding.
4.	The detailed analysis of error types—reasoning, vision recognition, knowledge, calculation, and question misunderstanding—provides useful insights into the model’s weaknesses and areas for improvement.

**Weaknesses:**

1.	While the dataset is innovative, the model’s architecture and training process primarily involve data engineering and fine-tuning rather than a new modeling approach. This diminishes the paper's novelty as it focuses on enhancing existing models with improved data rather than proposing a novel methodological framework.
2.	The paper does not provide in-depth information about adjustments made to the base model architectures (GLM-4V, CogVLM) to optimize them for mathematical reasoning tasks. Adding technical details on modifications could clarify how MathGLM-Vision addresses the unique challenges of mathematical problem-solving.
3.	Although the paper mentions experiments on general vision-language benchmarks, the analysis is brief, and performance drops in non-mathematical tasks. This may suggest a trade-off between specialization in mathematical tasks and general multi-modal understanding, a limitation for applications beyond mathematics.

**Questions:**

1.	In Section 2, the authors mention that MathVL includes “step-by-step solutions.” Were these solutions uniformly generated for all problem types, or was there variation based on the problem’s complexity? Could examples of different solution formats be provided?
2.	When curating MathVL, which specific properties were prioritized in selecting public datasets? Was the primary focus on dataset size, diversity of problem types, or specific visual elements?
3.	There is limited explanation on how the model handles equations or complex symbolic mathematics within visual inputs. For instance, were any specialized tokenization or embedding techniques employed for this type of input?
4.	Were any strategies, like curriculum learning, employed to ensure the model progressed from simpler to more complex mathematical problems? Adding this detail could be useful for readers interested in the training dynamics.
5.	MathGLM-Vision is designed to handle multiple images per problem, yet most benchmarks (e.g., MathVista) are single-image-based. How did the model perform on tasks involving single versus multiple images? Could the authors provide specific examples from MathVL-test where multiple images significantly contributed to?

---

### Meta-Review · Area_Chair_6QLR · 2024-12-22

**Metareview:**

This paper introduces a diverse and comprehensive multi-modal mathematical dataset MathVL, a benchmark dataset MathVL-test in a multi-image format, and several MLLMs of different scales fine-tuned on MathVL which show promising performance. While the reviewers appreciate these efforts, they pose a few concerns:  1. while the primary contribution is the MathVL dataset, there is no open-source plan to make it publicly available, so that others can replicate or build upon the proposed model; 2. the developed model is not state-of-the-art, and is essentially a fine-tuning version of the proposed dataset without any architectural innovation or modeling/training methodology innovation. No rebuttal is provided. AC suggests the authors address the above concerns and other questions posed by the reviewers.

**Additional Comments On Reviewer Discussion:**

The main concern raised by the reviewers:

the developed model is not the state-of-the-art, and is just a fine-tuning without any architectural innovation. And so the main contribution of this work is the dataset MathVL, which has no open-source plan.

No rebuttal is provided from the authors.

I agree with the reviewers that the dataset MathVL is the main contribution of this work. Without making it publicly available, its value to the community would be greatly diminished.

---

### Decision · Program_Chairs · 2025-01-22

Reject